# CATASTROPHIC OVERFITTING IS A BUG
# BUT IT IS CAUSED BY FEATURES

## ABSTRACT

Adversarial training (AT) is the *de facto* method to build robust neural networks, but it is computationally expensive. To overcome this, fast single-step attacks can be used, but doing so is prone to catastrophic overfitting (CO). This is when networks gain non-trivial robustness during the first stages of AT, but then reach a breaking point where they become vulnerable in just a few iterations. Although some works have succeeded at preventing CO, the different mechanisms that lead to this failure mode are still poorly understood. In this work, we study the onset of CO in single-step AT methods through controlled modifications of typical datasets of natural images. In particular, we show that CO can be *induced* when injecting the images with seemingly innocuous features that are very useful for non-robust classification but need to be combined with other features to obtain a robust classifier. This new perspective provides important insights into the mechanisms that lead to CO and improves our understanding of the general dynamics of adversarial training.

## 1 INTRODUCTION

Deep neural networks are sensitive to imperceptible worst-case perturbations, also known as *adversarial perturbations* (Szegedy et al., 2014). As a consequence, training neural networks that are robust to such perturbations has been an active area of study in recent years (see Ortiz-Jiménez et al. (2021) for a review). In particular, a prominent line of research, referred to as *adversarial training* (AT), focuses on online data augmentation with adversarial samples during training. However, it is well known that finding these adversarial samples for deep neural networks is an NP-hard problem (Weng et al., 2018). In practice, this is usually overcome with various methods, referred to as *adversarial attacks* that find approximate solutions to this hard problem. The most popular attacks are based on projected gradient descent (PGD) (Madry et al., 2018) – a computationally expensive algorithm that requires multiple steps of forward and backward passes through the neural network to approximate the solution. This hinders its use in many large-scale applications motivating the use of alternative efficient *single-step* attacks (Goodfellow et al., 2015; Shafahi et al., 2019; Wong et al., 2020).

The use of the computationally efficient single-step attacks within AT, however, comes with concerns regarding its stability. While training, although there is an initial increase in robustness, the networks often reach a breaking point beyond which they lose all gained robustness in just a few iterations (Wong et al., 2020). This phenomenon is known as *catastrophic overfitting* (CO) (Wong et al., 2020; Andriushchenko & Flammarion, 2020). Nevertheless, given the clear computational advantage of using single-step attacks during AT, a significant body of work has been dedicated to finding ways to circumvent CO via regularization and data augmentation (Andriushchenko & Flammarion, 2020; Vivek & Babu, 2020; Kim et al., 2021; Park & Lee, 2021; Golgooni et al., 2021; de Jorge et al., 2022).

Despite the recent methodological advances in this front, the *root cause of CO* remains poorly understood. Due to the inherent complexity of this problem, we argue that identifying the causal mechanisms behind CO cannot be done through observations alone and requires *active interventions* Ilyas et al. (2019). That is, we need to be able to synthetically induce CO in settings where it would not naturally happen otherwise.

In this work, we identify one such type of intervention that allows to perform abundant experiments to explain multiple aspects of CO. Specifically, the main contributions of our work are: (i) We show that CO can be induced by injecting features that, despite being strongly discriminative (i.e. useful for standard classification), are not sufficient for robust classification (see Fig. 1). (ii) Through

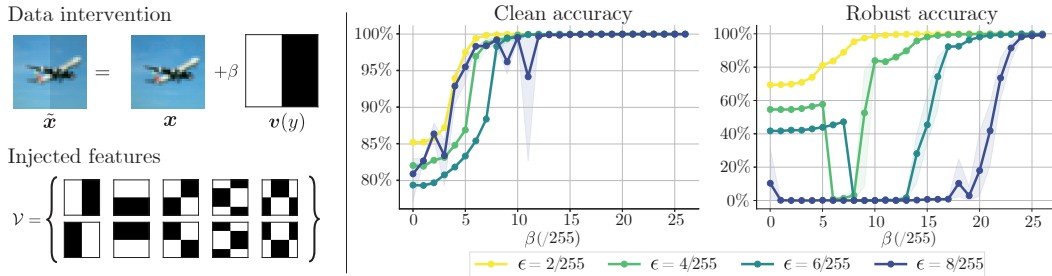

Figure 1: **Left:** Depiction of our modified dataset that injects simple, discriminative features. **Right:** Clean and robust performance after FGSM-AT on injected datasets $\widetilde{\mathcal{D}}_\beta$. We vary the strength of the synthetic features $\beta$ ($\beta = 0$ corresponds to the original CIFAR-10) and the robustness budget $\epsilon$ (train and test). We observe that for $\epsilon \in \{4/255, 6/255\}$ our intervention can induce CO when the synthetic features have strength $\beta$ slightly larger than $\epsilon$ while training on the original data does not suffer CO. Results are averaged over 3 seeds and shaded areas report minimum and maximum values.

extensive empirical analysis, we discover that CO is connected to the preference of the network to learn different features in a dataset. (iii) Building upon these insights, we describe and analyse a causal chain of events that can lead to CO. The main message of our paper is:

*Catastrophic overfitting is a learning shortcut used by the network to avoid learning complex robust features while achieving high accuracy using easy non-robust ones.*

Our findings improve our understanding of CO by focusing on how data influences AT. Moreover, they also provide insights in the dynamics of AT, in which the interaction between robust and non-robust features plays a key role.

**Outline** In Section 2, we give an overview of the related work on CO. Section 3 presents our main observation: CO can be induced by manipulating the data distribution. In Section 4, we perform an in-depth analysis of this phenomenon to identify the causes of CO. Finally, in Section 5 we use our new perspective to provide new insights on the different ways we can prevent CO.

## 2 PRELIMINARIES AND RELATED WORK

Let $f_{\boldsymbol{\theta}} : \mathbb{R}^d \to \mathcal{Y}$ denote a neural network architecture parameterized by a set of weights $\boldsymbol{\theta} \in \mathbb{R}^n$ which maps input samples $\boldsymbol{x} \in \mathbb{R}^d$ to $y \in \mathcal{Y} = \{1, \dots, c\}$. The objective of adversarial training (AT) is to find the network parameters $\boldsymbol{\theta} \in \mathbb{R}^n$ that optimize the following min-max problem:

$$\min_{\boldsymbol{\theta}} \mathbb{E}_{(\boldsymbol{x},y) \sim \mathcal{D}} \left[ \max_{\|\boldsymbol{\delta}\|_p \leq \epsilon} \mathcal{L}(f_{\boldsymbol{\theta}}(\boldsymbol{x} + \boldsymbol{\delta}), y) \right], \tag{1}$$

where $\mathcal{D}$ is some data distribution, $\boldsymbol{\delta} \in \mathbb{R}^d$ represents an adversarial perturbation, and $p, \epsilon$ characterize the adversary. This is typically solved by alternately minimizing the outer objective and maximizing the inner one via first-order optimization procedures. The outer minimization is tackled via some standard optimizer, *e.g.,* SGD, while the inner maximization problem is approximated with adversarial attacks like Fast Gradient Sign Method (FGSM) (Goodfellow et al., 2015) and Projected Gradient Descent (PGD) (Madry et al., 2018). Single-step AT methods are built on top of FGSM. In particular, FGSM solves the linearised version of the inner maximization objective. When $p = \infty$, this leads to:

$$\boldsymbol{\delta}_{\text{FGSM}} = \underset{\|\boldsymbol{\delta}\|_\infty \leq \epsilon}{\arg\max} \, \mathcal{L}(f_{\boldsymbol{\theta}}(\boldsymbol{x}), y) + \boldsymbol{\delta}^\top \nabla_{\boldsymbol{x}} \mathcal{L}(f_{\boldsymbol{\theta}}(\boldsymbol{x}), y) = \epsilon \, \text{sign}\left(\nabla_{\boldsymbol{x}} \mathcal{L}(f_{\boldsymbol{\theta}}(\boldsymbol{x}), y)\right). \tag{2}$$

Note that FGSM is very computationally efficient as it only requires a single forward-backward step. Unfortunately, FGSM-AT generally yields networks that are vulnerable to multi-step attacks such as PGD. In particular, Wong et al. (2020) observed that FGSM-AT presents a characteristic failure mode where the robustness of the model increases during the initial training epochs, but, at a certain point in training, the model loses all its robustness within the span of a few iterations. This is known as *catastrophic overfitting* (CO). They further observed that augmenting the FGSM attack with random noise seemed to mitigate CO. However, Andriushchenko & Flammarion (2020) showed that this method still leads to CO at larger $\epsilon$. Therefore, they proposed combining FGSM-AT with

a smoothness regularizer (GradAlign) that encourages the cross-entropy loss to be locally linear. Although GradAlign succeeds in avoiding CO in all tested scenarios, optimizing it requires the computation of a second-order derivative, which adds a significant computational overhead.

Several methods have been proposed that attempt to avoid CO while reducing the cost of AT. However these methods either only move CO to larger $\epsilon$ radii (Golgooni et al., 2021), are more computationally expensive (Shafahi et al., 2019; Li et al., 2020), or achieve sub-optimal robustness (Kang & Moosavi-Dezfooli, 2021; Kim et al., 2021). Recently, de Jorge et al. (2022) proposed N-FGSM that successfully avoids CO for large $\epsilon$ radii while incurring only a fraction of the computational cost of GradAlign.

On the more expensive side, multi-step attacks approximate the inner maximization in Eq. (1) with several gradient ascent steps (Kurakin et al., 2017; Madry et al., 2018; Zhang et al., 2019). Provided they use a sufficient number of steps, these methods do not suffer from CO and achieve better robustness. Nevertheless, using multi-step attacks in AT linearly increases the cost of training with the number of steps. Due to their superior performance and extensively validated robustness in the literature (Madry et al., 2018; Tramèr et al., 2018; Zhang et al., 2019; Rice et al., 2020) multi-step methods, such as PGD, are considered the reference in AT.

Aside from proposing methods to avoid CO, some works have also studied different aspects of the training dynamics when CO occurs. Wong et al. (2020) initially suggested that CO was a result of the networks overfitting to attacks limited to the corners of the $\ell_\infty$ ball. This conjecture was later dismissed by Andriushchenko & Flammarion (2020) who showed that AT with PGD attacks projected to the corners of the $\ell_\infty$ ball does not suffer from CO. Similarly, while Andriushchenko & Flammarion (2020) suggested that the reason that Wong et al. (2020) avoids CO is the reduced step size of the attack, de Jorge et al. (2022) showed they could prevent CO with noise augmentations while using a larger step-size. On the other hand, it has been consistently reported (Andriushchenko & Flammarion, 2020; Kim et al., 2021) that networks suffering from CO exhibit a highly non-linear loss landscape with respect to the input compared to their CO-free counterparts. As FGSM relies on the local linearity of the loss landscape, this sudden increase in non-linearity of the loss renders FGSM practically ineffective (Kim et al., 2021; Kang & Moosavi-Dezfooli, 2021). This provides a plausible explanation for why models are not fooled by FGSM after CO. However, none of these works have managed to identify what pushes the network to become strongly non-linear. In this work, we address this knowledge gap, and explore a plausible mechanism that can cause single-step AT to develop CO.

## 3 INDUCING CATASTROPHIC OVERFITTING

Our starting point is a well known observation: while robust solutions can be attained with non-trivial training procedures, *e.g.,* using AT, they are not the default consequence of standard training. That is, robust solutions are *harder to learn* and are avoided unless explicitly enforced (e.g. via adversarial training). On the other hand, we know robust classification requires leveraging alternative robust features that are not learned in the context of standard training (Ilyas et al., 2019; Sanyal et al., 2021), and, when CO happens, the robust accuracy plummets but the clean and FGSM accuracies do not drop; on the contrary, they keep increasing (Wong et al., 2020; Andriushchenko & Flammarion, 2020).

Bearing this in mind, we pose the following question: *Can CO be a mechanism to avoid learning the complex robust features?* If this is true, then the network could be using CO as a way to favour the learning of some very easy and non-robust features while ignoring the complex robust ones.

Directly testing this hypothesis, however, requires identifying and characterizing these two sets of features (robust *vs* non-robust) in a real dataset. This is a challenging task (as it is basically equivalent to solving the problem of adversarial robustness) that is beyond our current capabilities. For this reason, as is standard practice in the field (Arpit et al., 2017; Ilyas et al., 2019; Shah et al., 2020; Ortiz-Jimenez et al., 2020a), we take an alternative approach that relies on controlled modifications of the data. Conducting experiments on the manipulated data, we are able to make claims about CO and the structure of the data. In particular, we discover that if we inject a very simple discriminative feature on standard vision datasets then, under some conditions, we can induce CO at much lower values of $\epsilon$ than those for which it naturally happens without our intervention. This is a clear sign that the structure of the data plays a big role in the onset of CO.

**Our injected dataset** Let $(\boldsymbol{x}, y) \sim \mathcal{D}$ be an image-label pair sampled from a distribution $\mathcal{D}$. Our intervention modifies the *original* data $\boldsymbol{x}$ by adding an *injected* label-dependent feature $\boldsymbol{v}(y)$. We

construct a family of *injected datasets* $\widetilde{\mathcal{D}}_\beta$ where the label-dependent feature is scaled by $\beta$:

$$(\widetilde{\boldsymbol{x}}, y) \sim \widetilde{\mathcal{D}}_\beta : \quad \widetilde{\boldsymbol{x}} = \boldsymbol{x} + \beta\, \boldsymbol{v}(y) \quad \text{with} \quad (\boldsymbol{x}, y) \sim \mathcal{D}, \tag{3}$$

Moreover, we design $\boldsymbol{v}(y)$ such that $\|\boldsymbol{v}(y)\|_p = 1$ for all $y \in \mathcal{Y}$ and are linearly separable with respect to $y$. Since CO has primarily been observed for $\ell_\infty$ perturbations, we mainly use $p = \infty$ but also present some results with $p = 2$ in Appendix D. We denote the set of all injected features as $\mathcal{V} = \{\boldsymbol{v}(y) \mid y \in \mathcal{Y}\}$. The scale parameter $\beta > 0$ is fixed for all classes and controls the relative strength of the original and injected features, *i.e.,* $\boldsymbol{x}$ and $\boldsymbol{v}(y)$, respectively (see Fig. 1 (left)).

This construction has some interesting properties. Since the injected features are linearly separable and perfectly correlated with the labels, a linear classifier relying only on $\mathcal{V}$ can separate $\widetilde{\mathcal{D}}_\beta$ for a large enough $\beta$. Moreover, as $\beta$ also controls the classification margin, if $\beta \gg \epsilon$ this classifier is also robust. However, if $\boldsymbol{x}$ has some components in $\text{span}(\mathcal{V})$, the interaction between $\boldsymbol{x}$ and $\boldsymbol{v}(y)$ may decrease the robustness (or even clean accuracy) of the classifier for small $\beta$. We rigorously illustrate such a behaviour for linear classifiers in Appendix A. In short, although $\boldsymbol{v}(y)$ is easy-to-learn in general (we will empirically see that in Section 4.1), the discriminative power (and robustness) of a classifier that solely relies on the injected features $\mathcal{V}$ will depend on $\beta$.

With the aim to control the interactions between $x$ and $\boldsymbol{v}(y)$, we design $\mathcal{V}$ by selecting vectors from the low-frequency components of the 2D Discrete Cosine Transform (DCT) (Ahmed et al., 1974) as these have a large alignment with the space of natural images that we use for our experiments (*e.g.,* CIFAR-10). To ensure the norm constraint, we binarize these vectors so that they only take values in $\pm 1$, ensuring a maximal per-pixel perturbation that satisfies $\|\boldsymbol{v}(y)\|_\infty = 1$. These two design constraints also help to visually identify the alignment of adversarial perturbations $\boldsymbol{\delta}$ with $\boldsymbol{v}(y)$ as these patterns are visually distinctive (see Fig. 2).

**Injection strength ($\beta$) drives CO** We train a PreActResNet18 (He et al., 2016) on different intervened versions of CIFAR-10 (Krizhevsky & Hinton, 2009) using FGSM-AT for different robustness budgets $\epsilon$ and different scales $\beta$. Fig. 1 (right) shows a summary of these experiments both in terms of clean accuracy and robustness[1]. For the clean accuracy, Fig. 1 (right) shows two distinct regimes. First, when $\beta < \epsilon$, the network achieves roughly the same accuracy by training and testing on $\widetilde{\mathcal{D}}_\beta$ as by training and testing on $\mathcal{D}$ (corresponding to $\beta = 0$). This is expected as FGSM does not suffer from CO in this setting (see Fig. 1 (right)) and effectively ignores the added feature $\boldsymbol{v}(y)$. Meanwhile, when $\beta > \epsilon$, the clean test accuracy is almost $100\%$ (which is larger than

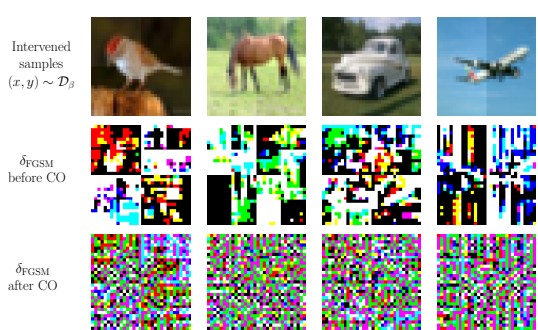

Figure 2: Different samples of the injected dataset $\widetilde{\mathcal{D}}_\beta$, and FGSM perturbations before and after CO. While prior to CO perturbations focus on the synthetic features, after CO they become noisy.

state-of-the-art accuracy on CIFAR-10) indicating that the network heavily relies on the injected features. We provide further evidence for this in Appendix E

The behaviour with respect to the robust accuracy is more interesting. For small $\epsilon$ ($\epsilon = 2/255$) the robust accuracy shows the same trend as the clean accuracy, albeit with lower values. For large $\epsilon$ ($\epsilon = 8/255$), the model incurs CO for most values of $\beta$. This is not surprising as CO has already been reported for this value of $\epsilon$ on the original CIFAR-10 dataset (Wong et al., 2020). However, the interesting setting is for intermediate values of $\epsilon$ ($\epsilon \in \{4/255, 6/255\}$). For these settings, Fig. 1 (right) distinguishes between three distinct regimes. The first two regimes are the same as for $\epsilon = 2/255$: (i) when the strength of the injected features is weak ($\beta \ll \epsilon$), the robust accuracy is similar to that trained on the original data ($\beta = 0$) and (ii) when it is strong ($\beta \gg \epsilon$), the robust accuracy is high as the network can use only $\boldsymbol{v}(y)$ to classify $\widetilde{\boldsymbol{x}}$ robustly. Nevertheless, there is a third regime where the injected features are mildly robust, *i.e.,* $\beta \approx \epsilon$. Strikingly, in this regime, the training suffers from CO and the robust accuracy drops to zero. This is significant, since training on the original dataset $\mathcal{D}$ ($\beta = 0$) does not suffer from CO for this value of $\epsilon$; but it does so when $\beta \approx \epsilon$.

---

[1]Throughout the paper, robustness is measured against strong PGD attacks with 50 iterations and 10 restarts.

We replicate these results for different $\mathcal{V}$'s and for $\ell_2$ perturbations with similar findings in Appendix D. Results for other datasets and networks, as well as further details of the training protocol are given in Appendices C and D respectively. From these observations, we conclude that there is indeed a link between the structure of the data and CO. In the following section, we delve deeper into these results to better understand the cause of CO.

## 4 ANALYSIS OF INDUCED CATASTROPHIC OVERFITTING

Since we now have a method to intervene in the data using Eq. (3) and induce CO, we can use it to better characterize the mechanisms that lead to CO. In particular, we explore how different features in the dataset influences the likelihood of observing CO in FGSM-AT.

### 4.1 ROBUST SOLUTIONS COMBINE EASY- AND HARD-TO-LEARN FEATURES

The previous section showed that when $\beta \ll \epsilon$ or $\beta \gg \epsilon$, our data intervention does not induce CO. However, for $\beta \approx \epsilon$ FGSM-AT consistently experiences CO. This begs the question: what makes $\beta \approx \epsilon$ special? We show that for $\beta \approx \epsilon$ a network trained using AT uses information from both the original dataset $\mathcal{D}$ and the injected features in $\mathcal{V}$ to achieve a high robust accuracy on the injected dataset $\widetilde{\mathcal{D}}_\beta$. However, when trained without any adversarial constraints *i.e.,* for standard training, the network only uses the features in $\mathcal{V}$ and achieves close to perfect clean accuracy.

In order to demonstrate this empirically, we perform standard, FGSM-AT, and PGD-AT training of a PreActResNet18 on the injected dataset $\widetilde{\mathcal{D}}_\beta$ (as described in Section 3) with $\beta = {}^8/_{255}$ and $\epsilon = {}^6/_{255}$. First, note that Fig. 1 (right) shows that an FGSM-AT model suffers from CO when trained on this injected dataset. Next, to identify the different features learned by these three models, we construct three different test sets and evaluate the clean and robust accuracy of the networks on them in Fig. 3. The three different test sets are: (i) CIFAR-10 test set with injected features ($\widetilde{\mathcal{D}}_\beta$), (ii) original CIFAR-10 test set ($\mathcal{D}$), and (iii) CIFAR-10 test set with shuffled injected features ($\widetilde{\mathcal{D}}_{\pi(\beta)}$) where the additive signals are correlated with a permuted set of labels, *i.e.,*

$$(\widetilde{\boldsymbol{x}}^{(\pi)}, y) \sim \widetilde{\mathcal{D}}_{\pi(\beta)}: \quad \widetilde{\boldsymbol{x}}^{(\pi)} = \boldsymbol{x} + \beta\,\boldsymbol{v}(\pi(y)) \quad \text{with} \quad (\boldsymbol{x}, y) \sim \mathcal{D} \quad \text{and} \quad \boldsymbol{v} \in \mathcal{V}. \tag{4}$$

Here, $\pi : \mathcal{Y} \to \mathcal{Y}$ is a fixed permutation operator that shuffles the labels. Note that evaluating these networks (trained on $\widetilde{\mathcal{D}}_\beta$) on data from $\widetilde{\mathcal{D}}_{\pi(\beta)}$ exposes them to contradictory information, since $\boldsymbol{x}$ and $\boldsymbol{v}(\pi(y))$ are correlated with different labels in $\widetilde{\mathcal{D}}_\beta$. Thus, if the classifier only relies on $\mathcal{V}$ the performance should be high on $\widetilde{\mathcal{D}}_\beta$ and low on $\mathcal{D}$ and $\widetilde{\mathcal{D}}_{\pi(\beta)}$, while if it only relies on $\mathcal{D}$ the performance should remain constant for all injected datasets.

**PGD training** We can conclude from Fig. 3(left) that the PGD-trained network achieves a robust solution using both $\mathcal{D}$ and $\mathcal{V}$. We know it uses $\mathcal{D}$ as it achieves better than trivial accuracy on $\mathcal{D}$ (containing no information from $\mathcal{V}$) as well as on $\widetilde{\mathcal{D}}_{\pi(\beta)}$ (where features from $\mathcal{V}$ are uncorrelated with the correct label; see Eq. (4)). On the other hand, we know the PGD-trained network uses $\mathcal{V}$ as the network achieves higher accuracy on $\widetilde{\mathcal{D}}_\beta$ (containing information from both $\mathcal{D}$ and $\mathcal{V}$) than $\mathcal{D}$ (only contains information from $\mathcal{D}$). Further, it suffers a drop in performance on $\widetilde{\mathcal{D}}_{\pi(\beta)}$ (where information from $\mathcal{D}$ and $\mathcal{V}$ are anti-correlated). This implies that the robust PGD solution effectively combines information from both the original and injected features for classification.

**Standard training** Standard training shows a completely different behaviour than PGD (see Fig. 3 (center)). In this case, even though the network achieves excellent clean accuracy on $\widetilde{\mathcal{D}}_\beta$, its accuracy on $\mathcal{D}$ is nearly trivial. This indicates that with standard training the model ignores the information present in $\mathcal{D}$ and only uses the non-robust features from $\mathcal{V}$ for classification. This is further supported by the observation that on $\widetilde{\mathcal{D}}_{\pi(\beta)}$, where the labels of the injected features are randomized, its accuracy is almost zero. From these observations we conclude that the injected features are easy to learn i.e. is preferred by our optimiser and neural networks.

**FGSM training** The behaviour of the FGSM-AT in Fig. 3(right) highlights the preference for the injected features even further. In the studied setting, FGSM-AT undergoes CO around epoch 8 when the robust accuracy on $\widetilde{\mathcal{D}}_\beta$ suddenly drops to zero despite a high clean accuracy on $\widetilde{\mathcal{D}}_\beta$. Next, as seen

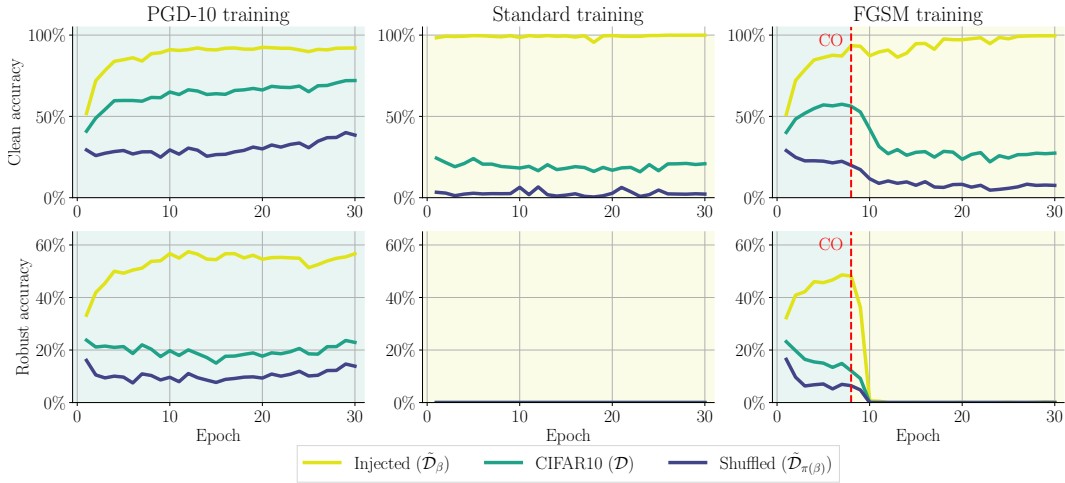

Figure 3: Clean (**top**) and robust (**bottom**) accuracy on 3 different test sets: (i) the original CIFAR-10 ($\mathcal{D}$), (ii) the dataset with injected features $\widetilde{\mathcal{D}}_\beta$ and (iii) the dataset with shuffled injected features $\widetilde{\mathcal{D}}_{\pi(\beta)}$. All training runs use $\beta = {}^8/_{255}$ and $\epsilon = {}^6/_{255}$ (where FGSM-AT suffers CO). The blue shading denotes when the network exploits both $\mathcal{D}$ and $\mathcal{V}$ and the yellow shading when it learns only $\mathcal{V}$.

in Fig. 3 (top right), FGSM-AT presents two distinct phases during training: (i) Prior to CO, when the robust accuracy on $\widetilde{\mathcal{D}}_\beta$ is non-zero, the network leverages features from both $\mathcal{D}$ and $\mathcal{V}$, similar to PGD. (ii) However, with the onset of CO, both the clean and robust accuracy on $\mathcal{D}$ and $\widetilde{\mathcal{D}}_{\pi(\beta)}$ drops, exhibiting behavior similar to standard training. This indicates that, post-CO, akin to standard training, the network forgets the information from $\mathcal{D}$ and solely relies on features in $\mathcal{V}$.

**Why does FGSM change the learned features after CO?** From the behaviour of standard training we concluded that the network has a preference for the injected features $\mathcal{V}$, *i.e.,* they are easy to learn. The behaviour of PGD training suggests that when the easy features are not sufficient to classify robustly, the model combines them with other (harder-to-learn) features *e.g.,* in $\mathcal{D}$, to become robust. FGSM initially learns a robust solution leveraging both $\mathcal{D}$ and $\mathcal{V}$ similar to PGD. However, if the FGSM attacks are rendered ineffective, the robustness constraints are essentially removed. This allows the network to revert back to the simple features and the performance on the original dataset $\mathcal{D}$ drops. This is, exactly, what occurs with the onset of CO around epoch 10 (further discussed in Section 4.2) Yet, why does CO happen in the first place? In the following, we will see that the key to answer this question lies in the way learning each type of feature influences the local geometry of the classifier.

## 4.2 CURVATURE EXPLOSION DRIVES CATASTROPHIC OVERFITTING

Recent works have shown that after the onset of CO the local geometry around the input $x$ becomes non-linear (Andriushchenko & Flammarion, 2020; Kim et al., 2021; de Jorge et al., 2022). Motivated by these findings, and with the aim to identify how our data intervention causes CO, we now investigate how the curvature of the loss landscape evolves with different types of training. We use the average maximum eigenvalue of the Hessian on $N = 100$ fixed training points $\bar{\lambda}_{\max} = \frac{1}{N}\sum_{n=1}^{N}\lambda_{\max}\left(\nabla_{\widetilde{x}}^2\mathcal{L}(f_\theta(\widetilde{x}_n), y_n)\right)$ to estimate the curvature, as suggested in Moosavi-Dezfooli et al. (2019)), and record it throughout training. Fig. 4(left) shows the result of this experiment for FGSM-AT (orange line) and PGD-AT (green line) training on $\widetilde{\mathcal{D}}_\beta$ with $\beta = {}^8/_{255}$ and $\epsilon = {}^6/_{255}$. Recall that this training regime exhibits CO with FGSM-AT around epoch 8 (see Fig. 3(left)).[2]

**Two-phase curvature increase** Interestingly, we observe that even before the onset of CO, both FGSM-AT and PGD-AT show a nearly similar steep increase in curvature (the $y$-axis is in logarithmic scale). While right before the $8^{th}$ epoch, there is a large increase in the curvature for PGD-AT, it stabilizes soon after and remains controlled for the rest of training. Prior work has observed that PGD-AT acts as a regularizer on the curvature (Moosavi-Dezfooli et al., 2019; Qin et al., 2019) which explains how PGD-AT controls the curvature explosion. However, curvature is a second-order

---

[2]Results for other parameters and for the original $\mathcal{D}$ are provided in Appendix F.

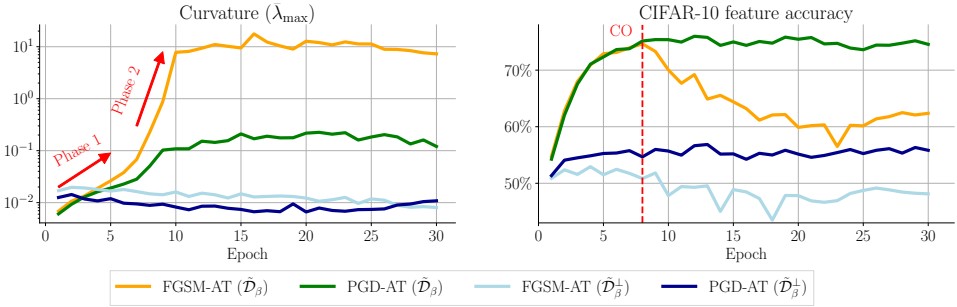

Figure 4: Evolution of different metrics for FGSM-AT and PGD-AT on 2 datasets: (i) with injected features ($\widetilde{\mathcal{D}}_\beta$) and (ii) with orthogonally projected features, i.e. with no interaction between the original and injected features ($\widetilde{\mathcal{D}}_\beta^\perp$). AT is performed for $\beta = {}^8/_{255}$ and $\epsilon = {}^6/_{255}$ (where FGSM suffers CO).

property of the loss surface and unlike PGD-AT, FGSM-AT is based on a coarse linear (first order) approximation of the loss. Therefore, FGSM-AT is not as effective at regularising the curvature. Indeed, we see that FGSM-AT cannot contain the curvature increase, which eventually explodes around the $8^{th}$ epoch and saturates at a very large value. Quite remarkably, the final curvature of the FGSM-AT model is 100 times that of the PGD-AT model.

**High curvature leads to meaningless perturbations** The fact that the curvature increases rapidly during CO, when the attack accuracy also increases, agrees with the findings of Andriushchenko & Flammarion (2020), that the loss becomes highly non-linear and thus reduces the success rate of FGSM. To show that CO indeed occurs due to the increased curvature breaking FGSM, we visualise the adversarial perturbations before and after CO. As observed in Fig. 2, before CO, the adversarial perturbations point in the direction of $\mathcal{V}$, albeit with some corruptions originating from $\boldsymbol{x}$. Nonetheless, after CO, the new adversarial perturbations point towards meaningless directions; they do not align with $\mathcal{V}$ even though the network is heavily reliant on this information for classifying the data (cf. Section 4.1). This reinforces the idea that the increase in curvature indeed causes a breaking point after which FGSM is no longer an effective adversarial attack. We would like to highlight that this behaviour of the adversarial perturbations after CO is radically different from the behaviour on standard and robust networks (in the absence of CO) where adversarial perturbations and curvature are strongly aligned with discriminative directions (Fawzi et al., 2018; Jetley et al., 2018; Ilyas et al., 2019).

### 4.3 CURVATURE INCREASE IS A RESULT OF INTERACTION BETWEEN FEATURES

But why does the network increase the curvature in the first place? In Section 4.1, we observed that this is a shared behaviour of PGD-AT and FGSM-AT, at least during the initial stage before CO. Therefore, it should not be a mere "bug". We conjecture that the curvature increases is a result of the interaction between features of the dataset which forces the network to increase its non-linearity in order to combine them effectively to obtain a robust model.

**Curvature does not increase without interaction** To demonstrate this, we perform a new experiment in which we modify $\mathcal{D}$ again (as in Section 3). However, this time, we ensure that there is no interaction between the synthetic features $\boldsymbol{v}(y)$ and the features from $\mathcal{D}$. We do so by creating $\widetilde{\mathcal{D}}_\beta^\perp$ such that:

$$(\widetilde{\boldsymbol{x}}^\perp, y) \sim \widetilde{\mathcal{D}}_\beta^\perp : \quad \widetilde{\boldsymbol{x}}^\perp = \mathcal{P}_{\mathcal{V}^\perp}(\boldsymbol{x}) + \beta \boldsymbol{v}(y) \quad \text{with} \quad (\boldsymbol{x}, y) \sim \mathcal{D} \quad \text{and} \quad \boldsymbol{v}(y) \in \mathcal{V} \tag{5}$$

where $\mathcal{P}_{\mathcal{V}^\perp}$ denotes the projection operator onto the orthogonal complement of $\mathcal{V}$. Since the synthetic features $\boldsymbol{v}(y)$ are orthogonal to $\mathcal{D}$, a simple linear classifier relying only on $\boldsymbol{v}(y)$ can robustly separate the data up to a radius that depends solely on $\beta$ (see the theoretical construction in Appendix A).

Interestingly, we find that, in this dataset, none of the $(\beta, \epsilon)$ configurations used in Fig. 5 induce CO. Here, we observe only two regimes: one that ignores $\mathcal{V}$ (when $\beta < \epsilon$) and one that ignores $\mathcal{D}$ (when $\beta > \epsilon$). This supports our conjecture that the interaction between the features of $\boldsymbol{x}$ and $\boldsymbol{v}(y)$ is the true cause of CO in $\widetilde{\mathcal{D}}_\beta$. Moreover, Fig. 4 (left) shows that, when performing either FGSM-AT (light blue) or PGD-AT (dark blue) on $\widetilde{\mathcal{D}}_\beta^\perp$, the curvature is consistently low. This agrees with the fact that in this case there is no need for the network to combine the injected and the original features to achieve robustness and hence the network does not need to increase its non-linearity to separate the data.

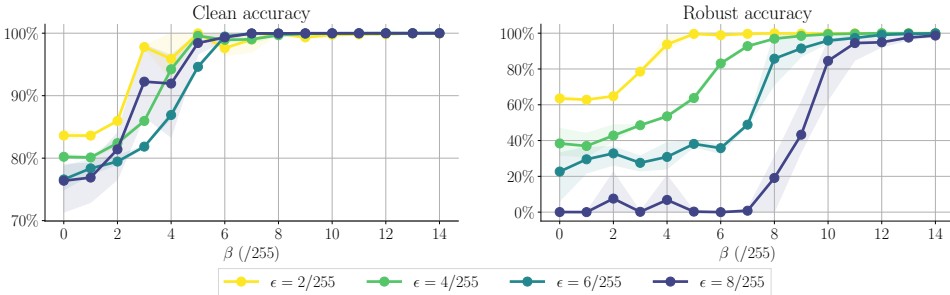

Figure 5: Clean (**left**) and robust (**right**) accuracy after FGSM-AT on a dataset with orthogonally injected features $\widetilde{\mathcal{D}}_\beta^\perp$ i.e. no interaction between original and injected features. We train while varying the strength of the injected features and the robustness budget $\epsilon$.

**Non-linear feature extraction** Finally, we study the relationship between the information learned in the features and the network's curvature. Using a similar methodology as Shi et al. (2022), we train multiple logistic classifiers on the the feature representations of $\mathcal{D}$ (output of the penultimate layer) of networks trained on $\widetilde{\mathcal{D}}_\beta$. Note that the accuracy of these classifiers strictly depends on how well the network (trained on $\widetilde{\mathcal{D}}_\beta$) has learned both $\mathcal{D}$ and $\mathcal{V}$. We will call this metric *feature accuracy*. Figure 4(right) shows the evolution of the feature accuracy of the networks during training. Observe that, for PGD-AT (green), the feature accuracy on $\mathcal{D}$ progressively grows during training. The high values of feature accuracy indicate that this network has learned to meaningfully extract information from $\mathcal{D}$, even if it was trained on $\widetilde{\mathcal{D}}_\beta$. Moreover, we note that the feature accuracy closely matches the curvature trajectory in Fig. 4 (left). Meanwhile, for FGSM-AT the feature accuracy has two phases: first, it grows at the same rate as for the PGD-AT network, but when CO happens, it starts to decrease. Note, however, that the curvature does not decrease. We argue this is because the network is using a shortcut to ignore $\mathcal{D}$. Specifically, if the curvature is very high, FGSM is rendered ineffective and allows the network to focus only on the easy non-robust features. On the other hand, if we use the features from networks trained on $\widetilde{\mathcal{D}}_\beta^\perp$ we observe that the accuracy on $\mathcal{D}$ is always low. This reinforces the view that the network is increasing the curvature in order to improve its feature representation. In $\widetilde{\mathcal{D}}_\beta^\perp$ the network does not need to combine information from both $\mathcal{D}$ and $\mathcal{V}$ to become robust, and hence it does not learn to extract information from $\mathcal{D}$.

### 4.4 A MECHANISTIC EXPLANATION OF CO

To summarize, we finally describe the chain of events that leads to CO in our injected datasets:

 (i) To learn a robust solution, the network attempts to combine easy, non-robust features with more complex robust features. However, without robustness constraints, the network strongly favors learning *only* the non-robust features (see Section 4.1).

 (ii) When learning both kinds of features simultaneously, the network increases its non-linearity to improve its feature extraction ability (see Section 4.3).

(iii) This increase in non-linearity provides a shortcut to break FGSM which triggers CO. This allows the network to avoid learning the complex robust features while still achieving a high accuracy using only the easy non-robust ones (see Section 4.2).

Aside from our empirical study, the intuition that a classifier needs to combine features to become robust can be formalized in certain settings. For example, in Appendix B we mathematically prove that there exist many learning problems in which learning a robust classifier requires leveraging additional non-linear features on top of the simple ones used for the clean solution. Moreover, in Section 5 we leverage these intuitions to explore how data interventions on real datasets can prevent CO.

## 5 FURTHER INSIGHTS AND DISCUSSION

Our proposed dataset intervention, defined in Section 3, allowed us to gain a better understanding of the chain of events that lead to CO. In this section, we focus our attention on methods that can prevent CO and analyze them in the context of our framework to provide further insights.

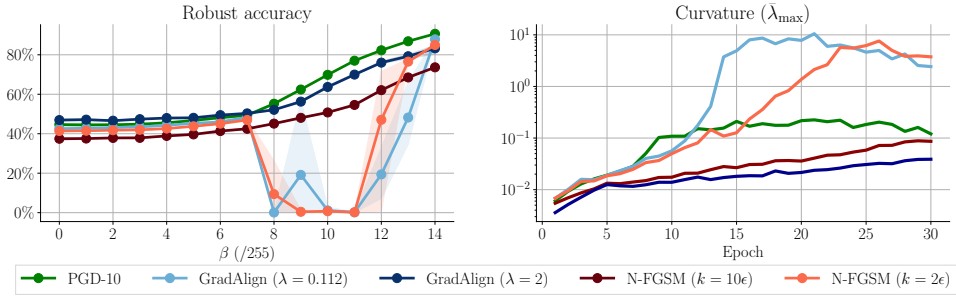

Figure 6: **Left:** Clean and robust performance after AT with GradAlign, N-FGSM and PGD-10 on $\widetilde{\mathcal{D}}_\beta$ at $\epsilon = 6/255$. Results averaged over three random seeds and shaded areas report minimum and maximum values. **Right:** Curvature evolution when training on $\widetilde{\mathcal{D}}_\beta$ at $\epsilon = 6/255$ and $\beta = 8/255$.

**GradAlign and N-FGSM prevent CO on $\widetilde{\mathcal{D}}_\beta$** In Fig. 6, we show the robust accuracy (left) and curvature (right) of models trained with GradAlign (Andriushchenko & Flammarion, 2020) and N-FGSM (de Jorge et al., 2022) on $\widetilde{\mathcal{D}}_\beta$ for varying $\beta$. Figure 6 (left) shows that both methods are able to prevent CO on our injected dataset for suitable choices of the regularisation parameter *i.e.,* $\lambda$ for GradAlign and $k$ for N-FGSM. This suggests that the mechanism by which our intervention induces CO is similar to how it occurs in real datasets. However, for certain values of $\beta$, both GradAlign and N-FGSM require stronger regularization. Thus, the regularization strength is not only a function of $\epsilon$, as discussed in their respective manuscripts, but also of the signal strength $\beta$. As $\beta$ increases, $\boldsymbol{v}(y)$ becomes more discriminative creating a stronger incentive for the network to use it. We argue that this increases the chances for CO as the network (based on our observations in Section 4) will likely increase the curvature to combine the discriminative injected features with others in order to become robust. Moreover, Fig. 6 shows that the curvature of N-FGSM and GradAlign AT stabilizes with a trend similar to PGD-AT and stronger regularizations dampens the increase of the curvature even further. This further shows that preventing the curvature from exploding can indeed prevent CO.

**Can data modifications avoid CO?** Section 3 shows that it is possible to induce CO through data manipulations. But is the opposite also true that CO can be avoided using data manipulations? We find that this is indeed possible on CIFAR-10. Table 1 shows that removing the high frequency components of $\mathcal{D}$ consistently prevents CO at $\epsilon = 8/255$ (where FGSM-AT fails). Interestingly, Grabinski et al. (2022), in concurrent work, applied this idea to the pooling layers and showed they could prevent CO. Surprisingly, though, we have found that applying the same low-pass technique at $\epsilon = 16/255$ does not work. We conjecture this is because the features which are robust at $\epsilon = 8/255$

Table 1: Clean and robust accuracies of FGSM-AT and PGD-AT trained networks on CIFAR-10 and the low pass version described in Ortiz-Jimenez et al. (2020b) at different $\epsilon$.

| Method ($\epsilon$) | Original | | Low pass | |
|---|---|---|---|---|
| | Clean | Robust | Clean | Robust |
| FGSM ($8/255$) | 85.6 | 0.0 | 81.1 | 47.0 |
| PGD ($8/255$) | 80.9 | 50.6 | 80.3 | 49.7 |
| FGSM ($16/255$) | 76.3 | 0.0 | 78.6 | 0.0 |
| PGD ($16/255$) | 68.0 | 29.2 | 66.9 | 28.4 |

in the low pass version of CIFAR-10 might not be robust at $\epsilon = 16/255$, therefore forcing the network to combine more complex features. Although this method does not work in all settings, it is an interesting proof of concept that shows that removing some features from the dataset can indeed prevent CO. Generalizing this idea to other kinds of features and datasets can be a promising avenue for future work.

## 6    CONCLUDING REMARKS

In this work, we have presented a thorough empirical study to establish a causal link between the features of the data and the onset of CO in FGSM-AT. Specifically, using controlled data interventions we have seen that catastrophic overfitting is a learning shortcut used by the network to avoid learning hard-to-learn robust features while achieving high accuracy using easy non-robust ones. This new perspective has allowed us to shed new light on the mechanisms that trigger CO, as it shifted our focus towards studying the way the data structure influences the learning algorithm. We believe this opens the door to promising future work focused on understanding the intricacies of these learning mechanisms. In general, we consider that deriving methods for inspecting the data and identifying how different features of a dataset interact with each other as in Ilyas et al. (2019) is another interesting avenue for future work.

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

## A  ANALYSIS OF THE SEPARABILITY OF THE INJECTED DATASETS

With the aim to illustrate how the interaction between $\mathcal{D}$ and $\mathcal{V}$ can influence the robustness of a classifier trained on $\widetilde{\mathcal{D}}_\beta$ we now provide a toy theoretical example in which we discuss this interaction. Specifically, without loss of generality, consider the binary classification setting on the dataset $(\boldsymbol{x}, y) \sim \mathcal{D}$ where $y \in \{-1, +1\}$ and $\|\boldsymbol{x}\|_2 = 1$, for ease. Let's now consider the injected dataset $\widetilde{\mathcal{D}}_\beta$ and further assume that $\boldsymbol{v}(+1) = \boldsymbol{u}$ and $\boldsymbol{v}(-1) = -\boldsymbol{u}$ with $\boldsymbol{u} \in \mathbb{R}^d$ and $\|\boldsymbol{u}\|_2 = 1$, such that $\widetilde{\boldsymbol{x}} = \boldsymbol{x} + \beta y \boldsymbol{u}$. Moreover, let $\gamma \in [0, 1]$ denote the *interaction coefficient* between $\mathcal{D}$ and $\mathcal{V}$, such that $-\gamma \le \boldsymbol{x}^\top \boldsymbol{u} \le \gamma$.

We are interested in characterizing the robustness of a classifier that only uses information in $\mathcal{V}$ when classifying $\widetilde{\mathcal{D}}_\beta$ depending on the strength of the interaction coefficient. In particular, as we are dealing with the binary setting, we will characterize the robustness of a linear classifier $h : \mathbb{R}^d \to \{-1, +1\}$ that discriminates the data based only on $\mathcal{V}$, *i.e.,* $h(\widetilde{\boldsymbol{x}}) = \text{sign}(\boldsymbol{u}^\top \widetilde{\boldsymbol{x}})$. In our setting, we have

$$\boldsymbol{u}^\top \widetilde{\boldsymbol{x}} = \boldsymbol{u}^\top \boldsymbol{x} + \beta \boldsymbol{u}^\top \boldsymbol{u} = \boldsymbol{u}^\top \boldsymbol{x} + \beta \quad \text{if } y = +1$$
$$\boldsymbol{u}^\top \widetilde{\boldsymbol{x}} = \boldsymbol{u}^\top \boldsymbol{x} - \beta \boldsymbol{u}^\top \boldsymbol{u} = \boldsymbol{u}^\top \boldsymbol{x} - \beta \quad \text{if } y = -1$$

**Proposition 1** (Clean performance). *If $\beta > \gamma$, then $h$ achieves perfect classification accuracy on $\widetilde{\mathcal{D}}_\beta$.*

*Proof.* Observe that if $\gamma = 0$, i.e. the features from original dataset $\mathcal{D}$ do not interact with the injected features $\mathcal{V}$, the dataset is perfectly linearly separable. However, if the data $\boldsymbol{x}$ from $\mathcal{D}$ interacts with the injected signal $\boldsymbol{u}$, i.e. non zero projection, then the dataset is still perfectly separable but for a sufficiently larger $\beta$, such that $\boldsymbol{u}^\top \boldsymbol{x} + \beta > 0$ when $y = +1$ and $\boldsymbol{u}^\top \boldsymbol{x} + \beta < 0$ when $y = -1$. Because $-\gamma \le \boldsymbol{x}^\top \boldsymbol{u} \le \gamma$ this is achieved for $\beta > \gamma$. $\square$

**Proposition 2** (Robustness). *If $\beta > \gamma$, the linear classifier $h$ is perfectly accurate and robust to adversarial perturbations in an $\ell_2$-ball of radius $\epsilon \le \beta - \gamma$. Or, equivalently, for $h$ to be $\epsilon$-robust, the injected features must have a strength $\beta \ge \epsilon + \gamma$.*

*Proof.* Given $\widetilde{\boldsymbol{x}}$, we seek to find the minimum distance to the decision boundary of such a classifier. A minimum distance problem can be cast as solving the following optimization problem:

$$\epsilon^\star(\widetilde{\boldsymbol{x}}) = \min_{\boldsymbol{r} \in \mathbb{R}^d} \|\boldsymbol{r} - \widetilde{\boldsymbol{x}}\|_2^2 \quad \text{subject to } \boldsymbol{r}^\top \boldsymbol{u} = 0,$$

which can be solved in closed form

$$\epsilon^\star(\widetilde{\boldsymbol{x}}) = \frac{|\boldsymbol{u}^\top \widetilde{\boldsymbol{x}}|}{\|\boldsymbol{u}\|} = |\boldsymbol{u}^\top \boldsymbol{x} + y\beta|.$$

The robustness radius of the classifier $h$ will therefore be $\epsilon = \inf_{\widetilde{\boldsymbol{x}} \in \text{supp}(\widetilde{\mathcal{D}}_\beta)} \epsilon^\star(\widetilde{\boldsymbol{x}})$, which in our case can be bounded by

$$\epsilon = \inf_{(\widetilde{\boldsymbol{x}}, y) \in \text{supp}(\widetilde{\mathcal{D}}_\beta)} \epsilon^\star(\widetilde{\boldsymbol{x}}) \le \min_{|\boldsymbol{u}^\top \boldsymbol{x}| \le \gamma, y = \pm 1} |\boldsymbol{u}^\top \boldsymbol{x} + y\beta| = |\mp \gamma \pm \beta| = \beta - \gamma.$$

$\square$

Based on these propositions, we can clearly see that the interaction coefficient $\gamma$ reduces the robustness of the additive features $\mathcal{V}$. In this regard, if $\epsilon \ge \beta - \gamma$, robust classification at a radius $\epsilon$ can only be achieved by also leveraging information within $\mathcal{D}$.

## B  ROBUST CLASSIFICATION CAN REQUIRE NON-LINEAR FEATURES

We now provide a rigorous theoretical example of a learning problem that provably requires additional complex information for robust classification, even though it can achieve good clean performance using only simple features.

Given some $p \in \mathbb{N}$, let $\mathbb{R}^{p+1}$ be the input domain. A concept class, defined over $\mathbb{R}^{p+1}$ is a set of functions from $\mathbb{R}^{p+1}$ to $\{0, 1\}$. A hypothesis $h$ is $s$-non-linear if the polynomial with the smallest degree that can represent $h$ has a degree (largest order polynomial term) of $s$.

Using these concepts we now state the main result.

**Theorem 1.** *For any $p, k \in \mathbb{N}, \epsilon < 0.5$ such that $k < p$, there exits a family of distributions $\mathcal{D}_k$ over $\mathbb{R}^{p+1}$ and a concept class $\mathcal{H}$ defined over $\mathbb{R}^{p+1}$ such that*

1. *$\mathcal{H}$ is PAC learnable (with respect to the clean error) with a* linear (degree 1) *classifier. However, $\mathcal{H}$ is not robustly learnable with any linear classifier.*

2. *There exists an efficient learning algorithm, that given a dataset sampled i.i.d. from a distribution $\mathcal{D} \in \mathcal{D}_k$ robustly learns $\mathcal{H}$.*

*In particular, the algorithm returns a $k$-non-linear classifier and in addition, the returned classifier also exploits the linear features used by the linear non-robust classifier.*

*Proof.* We now define the construction of the distributions in $\mathcal{D}_k$. Every distribution $\mathcal{D}$ in the family of distribution $\mathcal{D}_k$ is uniquely defined by three parameters: a threshold parameter $\rho \in \{4t\epsilon : t \in \{0, \cdots, k\}\}$ (one can think of this as the non-robust, easy-to-learn feature), a $p$ dimensional bit vector $\boldsymbol{c} \in \{0, 1\}^p$ such that $\|\boldsymbol{c}\|_1 = k$ (this is the non-linear but robust feature) and $\epsilon$. Therefore, given $\rho$ and $\boldsymbol{c}$ (and $\epsilon$ which we discuss when necessary and ignore from the notation for simplicity), we can define the distribution $\mathcal{D}^{\boldsymbol{c}, \rho}$. We provide an illustration of this distribution for $p = 2$ in Figure 7.

**Sampling the robust non-linear feature**  To sample a point $(\boldsymbol{x}, y) \in \mathbb{R}^{p+1}$ from the distribution $\mathcal{D}^{\boldsymbol{c}, \rho}$, first, sample a random bit vector $\hat{\boldsymbol{x}} \in \mathbb{R}^p$ from the uniform distribution over the boolean hypercube $\{0, 1\}^p$. Let $\hat{y} = \sum_{i=1}^{p-1} \hat{\boldsymbol{x}}[i] \cdot \boldsymbol{c}[i] \pmod 2$ be the label of the parity function with respect to $\boldsymbol{c}$ evaluated on $\hat{\boldsymbol{x}}$. The marginal distribution over $\hat{y}$, if sampled this way, is equivalent to the Bernoulli distribution with parameter $\frac{1}{2}$. To see why, fix all bits in the input except one (chosen arbitrarily from the variables of the parity function), which is distributed uniformly over $\{0, 1\}$. It is easy to see that this forces the output of the parity function to be distributed uniformly over $\{0, 1\}$ as well. Repeating this process for all dichotomies of $p - 1$ variables of the parity function proves the desired result. Intuitively, $\hat{\boldsymbol{x}}$ constitutes the robust non-linear feature of this distribution.

**Sampling the non-robust linear feature**  To ensure that $\hat{\boldsymbol{x}}$ is not perfectly correlated with the true label, we sample the true label $y$ from a Bernoulli distribution with parameter $\frac{1}{2}$. Then we sample the non-robust feature $\boldsymbol{x}_1$ as follows

$$\boldsymbol{x}_1 \sim \begin{cases} \text{Unif} \left(X_1^-\right) & y = 0 \wedge \hat{y} = 0 \\ \text{Unif} \left(X_1^+\right) & y = 1 \wedge \hat{y} = 1 \\ \text{Unif} \left(X_2^-\right) & y = 0 \wedge \hat{y} = 1 \\ \text{Unif} \left(X_2^+\right) & y = 1 \wedge \hat{y} = 0 \end{cases}$$

where

$$X_1^+ = [\rho, \rho + \epsilon] \quad \text{and} \quad X_2^+ = [(\rho + 2\epsilon, \rho + 3\epsilon)], X_1^- = [\rho - \epsilon, \rho] \quad \text{and} \quad X_2^- = [(\rho - 3\epsilon, \rho - 2\epsilon)].$$

Finally, we return $(\boldsymbol{x}, y)$ where $\boldsymbol{x} = (\boldsymbol{x}_1; \hat{\boldsymbol{x}})$ is the concatenation of $\boldsymbol{x}_1$ and $\hat{\boldsymbol{x}}$.

**Linear non-robust solution**  First, we show that there is a linear, accurate, but non-robust solution to this problem. To obtain this solution, sample an $m$-sized dataset $S_m = \{(\boldsymbol{x}_1, y_1), \ldots, (\boldsymbol{x}_m, y_m)\} \in \mathbb{R}^{p+1} \times \{-1, 1\}$ from the distribution $\mathcal{D}^{\boldsymbol{c}, \rho}$. Ignore all, except the first coordinate, of the covariates of the dataset to create $S_m^0 = \{(\boldsymbol{x}_1 [0], y_1) \ldots (\boldsymbol{x}_m [0], y_m)\}$ where $S_i [j]$ indexes the $j^{th}$ coordinate of the $i^{th}$ element of the dataset. Then, sort $S_m^0$ on the basis of the covariates (*i.e.,* the first coordinate). Let $\hat{\rho}$ be the largest element whose label is 0.

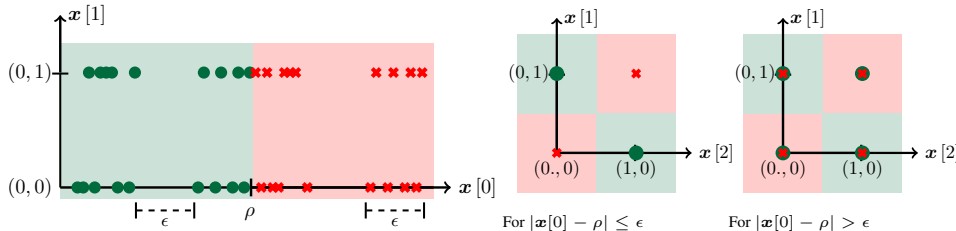

Figure 7: Illustration of one possible distribution $\mathcal{D}^{c,\rho}$ in three dimensions. The data is linearly separable in the direction $x[0]$ but has a very smalll margin in that direction. Leveraging $x[1]$ and $x[2]$ additionally, we see how the data can indeed be separated robustly, albeit non-linearly.

Define $f_{\text{lin},\hat{\rho}}$ as the linear threshold function on the first coordinate i.e. $f_{\text{lin},\hat{\rho}}(x) = \mathbb{I}\{x[0] \geq \hat{\rho}\}$. By construction, $f_{\text{lin},\hat{\rho}}$ accurate classifies all points in $S_m$. The VC dimension of a linear threshold function in 1 dimension is 2. Then, using standard VC sample complexity upper bounds [3] for consistent learners, if $m \geq \kappa_0 \left(\frac{1}{\alpha} \log\left(\frac{1}{\beta}\right) + \frac{1}{\alpha} \log\left(\frac{1}{\alpha}\right)\right)$, where $\kappa_0$ is some universal constant, we have that with probability at least $1 - \beta$,

$$\text{Err}\left(f_{\text{lin},\hat{\rho}}; \mathcal{D}^{c,\rho}\right) \leq \alpha.$$

**Non-linear robust solution**   Next, we propose an algorithm to find a robust solution and show that this solution has a non-linearity of degree $k$. First, sample the $m$-sized dataset $S_m$ and use the method described above to find $\hat{\rho}$. Then, create a modified dataset $\widehat{S}$ by first removing all points $x$ from $S_m$ such that $|x[0] - \hat{\rho}| \geq \frac{\epsilon}{8}$ and then removing the first coordinate of the remaining points. Thus, each element in $\widehat{S}$ belongs to $\mathbb{R}^p \times \{0, 1\}$ dimensional dataset.

Note, that by construction, there is a consistent (*i.e.,* an accurate) parity classifier on $\widehat{S}$. Let the parity bit vector consistent with $\widehat{S}$ be $\hat{c}$. This can be learned using Gaussian elimination. Consequently, construct the parity classifier $f_{\text{par},\hat{c}} = \sum_{i=0}^{p-1} \widehat{x}[i] \cdot \widehat{c}[i] \pmod 2$.

Finally, the algorithm returns the classifier $g_{\hat{\rho},\hat{c}}$, which acts as follows:

$$g_{\hat{\rho},\hat{c}}(x) = \begin{cases} 1 & \mathbb{I}\left\{x[0] \geq \hat{\rho} + \epsilon + \frac{\epsilon}{8}\right\} \\ 0 & \mathbb{I}\left\{x[0] \leq \hat{\rho} - \epsilon - \frac{\epsilon}{8}\right\} \\ f_{\text{par},\hat{c}}(\widetilde{x}) & \text{o.w.} \end{cases} \tag{6}$$

where $\widetilde{x} = \text{round}(x[1, \ldots, p])$ is obtained by rounding off $x$ starting from the second index till the last. For example, if $x = [0.99, 0.4, 0.9, 0.4, 0.8]$, $\epsilon = 0.2$, and $c = [0, 0, 1, 1]$ then $\widetilde{x} = [0, 1, 0, 1]$ and $g_{0.5,\hat{c}}[\widetilde{x}] = 1$. Finally, it is easy to verify that the classifier $g_{\hat{\rho},\hat{c}}$ is accurate on all training points and as the number of total parity classifiers is less than $2^p$ (hence finite VC dimension), as long as $m \geq \kappa_1 \left(\frac{1}{\alpha} \log\left(\frac{1}{\beta}\right) + \frac{p}{\alpha} \log\left(\frac{1}{\alpha}\right)\right)$, where $\kappa_1$ is some universal constant, we have that with probability at least $1 - \beta$,

$$\text{Err}\left(g_{\hat{\rho},\hat{c}}; \mathcal{D}^{c,\rho}\right) \leq \alpha.$$

**Robustness of $g_{\hat{\rho},\hat{c}}$**   As $x[0]$ is distributed uniformly in the intervals $[\rho - \epsilon, \rho] \cup [\rho, \rho + \epsilon]$, we have that $|\rho - \hat{\rho}| \leq 4\epsilon \cdot \text{Err}\left(f_{\text{lin},\hat{\rho}}; \mathcal{D}^{c,\rho}\right) \leq 4\epsilon\alpha$. Therefore, when $m$ is large enough $\left(m = \text{poly}\left(\frac{1}{\epsilon}\right)\right)$ such that $\alpha \leq \frac{1}{32}$, we have that $|\hat{\rho} - \rho| \leq \frac{\epsilon}{8}$. Intuitively, this guarantees that $g_{\hat{\rho},\hat{c}}$ uses the linear threshold function on $x[0]$ for classification in the interval $[\rho, \rho + \epsilon] \cup [\rho, \rho - \epsilon]$ and $f_{\text{par},\hat{c}}$ in the $[\rho + 2\epsilon, \rho + 3\epsilon] \cup [\rho - 2\epsilon, \rho - 3\epsilon]$. A crucial property of $g_{\hat{\rho},\hat{c}}$ is that for all $x \in \text{Supp}\left(\mathcal{D}^{c,\rho}\right)$, the classifier $g_{\hat{\rho},\hat{c}}$ does not alter its prediction in an $\ell_\infty$-ball of radius $\epsilon$. We show this by studying four separate cases. First, we prove robustness along all coordinates except the first.

1. When $|x[0] - \hat{\rho}| \geq \epsilon + \frac{\epsilon}{8}$, as shown above, $g_{\hat{\rho},\hat{c}}$ is invariant to all $x[i]$ for all $i > 0$ and is thus robust against all $\ell_\infty$ perturbations against those coordinates.

---

[3] https://www.cs.ox.ac.uk/people/varun.kanade/teaching/CLT-MT2018/lectures/lecture03.pdf

2. When $|\boldsymbol{x}[0] - \hat{\rho}| < \epsilon + \frac{\epsilon}{8}$, due to Equation (6), we have that $g_{\hat{\rho},\hat{c}} = f_{\mathrm{par},\hat{c}}(\widetilde{\boldsymbol{x}})$ where $\widetilde{\boldsymbol{x}} = \mathrm{round}\,(\boldsymbol{x}\,[1,\ldots,p])$ is obtained by rounding off all indices of $\boldsymbol{x}$ except the first. As the rounding operation on the boolean hypercube is robust to any $\ell_\infty$ perturbation of radius less than 0.5, we have that $g_{\hat{\rho},\hat{c}}$ is robust to all $\ell_\infty$ perturbations of radius less than 0.5 on the support of the distribution $\mathcal{D}^{c,\rho}$.

Next, we prove the robustness along the first coordinate. Let $0 < \delta < \epsilon$ represent an adversarial perturbation. Without loss of generality, assume that $\boldsymbol{x}\,[0] > \hat{\rho}$ as similar arguments apply for the other case.

1. Consider the case $\boldsymbol{x}[0] \leq \hat{\rho} + \epsilon + \frac{\epsilon}{8}$. Then, $|\boldsymbol{x}\,[0] - \delta - \hat{\rho}| \leq \left|\epsilon + \frac{\epsilon}{8} - \delta\right| \leq \epsilon + \frac{\epsilon}{8}$ and hence, by construction, $g_{\hat{\rho},\hat{c}}(\boldsymbol{x}) = g_{\hat{\rho},\hat{c}}([x\,[0] - \delta; [x]\,[1,\ldots,p]])$. On the other hand, for all $\delta$, we have that $g_{\hat{\rho},\hat{c}}([x\,[0] + \delta; [x]\,[1,\ldots,p]]) = 1$ if $g_{\hat{\rho},\hat{c}}(x) = 1$.

2. For the case $\boldsymbol{x}[0] \geq \hat{\rho} + \epsilon + \frac{\epsilon}{8}$, the distribution is supported only on the interval $[\rho + 2\epsilon, \rho + 3\epsilon]$. When a positive $\delta$ is added to the first coordinate, the classifier's prediction does not change and it remains 1. For all $\delta \leq \frac{\epsilon}{2}$, when the perturbation is subtracted from the first coordinate, its first coordinate is still larger than $\hat{\rho} + \epsilon + \frac{\epsilon}{8}$ and hence, the prediction is still 1.

This completes the proof of robustness of $g_{\hat{\rho},\hat{c}}$ along all dimensions to $\ell_\infty$ perturbations of radius less than $\epsilon$. Combining this with its error bound, we have that $\mathrm{Adv}_{\epsilon,\infty}\,(g_{\hat{\rho},\hat{c}}; \mathcal{D}^{c,\rho}) \leq \alpha$.

To show that the parity function is non-linear, we use a classical result from Aspnes et al. (1994). Theorem 2.2 in Aspnes et al. (1994) shows that approximating the parity function in $k$ bits using a polynomial of degree $\ell$ incurs at least $\sum_{i=0}^{k_\ell} \binom{k}{i}$ where $k_\ell = \left\lfloor \frac{k-\ell-1}{2} \right\rfloor$ mistakes. Therefore, the lowest degree polynomial that can do the approximation accurately is at least $k$.

This completes our proof of showing that the robust classifier is of non-linear degree $k$ while the accurate classifier is linear. Next, we prove that no linear classifier can be robust. We show this by contradiction.

**No linear classifier can be robust**  Construct a set $\mathcal{Z}$ of $s$ (to be defined later) points in $\mathbb{R}^{p+1}$ by sampling the first coordinate from the interval $[\rho, \rho + \epsilon]$ and the remaining $p$ coordinates uniformly from the boolean hypercube. Then, augment the set by subtracting $\epsilon$ from the first coordinate while retaining the rest of the coordinates. Note that this set can be generated, along with its labels, by sampling enough points from the original distribution and discarding points that do not fall in this interval. Now construct adversarial examples of each point in the augmented set by either adding or subtracting $\epsilon$ from the negatively and the positively labelled examples respectively and augment the original set with these adversarial points. For a large enough $s$,[4] this augmented set of points can be decomposed into a multiset of points, where all points in any one set has the same value in the first coordinate but nearly half of their label is zero and the other half one.

Now, assume that there is a linear classifier that has a low error on the distribution $\mathcal{D}^{c,\rho}$. Therefore the classifier is also accurate on these sets of points as the classifier is robust, by assumption, and the union of these sets occupy a significant under the distribution $\mathcal{D}^{c,\rho}$. However, as the first coordinate of every point within a set is constant despite half the points having label one and the other half zero, the coefficient of the linear classifier can be set to zero without altering the behavior of the classifier. Then, effectively the linear classifier is representing a parity function on the rest of the $p$ coordinates. However, we have just seen that this is not possible as a linear threshold function cannot represent a parity function on $k$ bits where $k > 1$. This contradicts our initial assumption that there is a robust linear classifier for this problem.

This completes the proof. □

---

[4]There is a slight technicality as we might not obtain points that are exact reflections of each other around $\rho$ but that can be overcome by discretising upto a certain precision

## C  EXPERIMENTAL DETAILS

In this section we provide the experimental details for all results presented in the paper. Adversarial training for all methods and datasets follows the fast training schedules with a cyclic learning rate introduced in Wong et al. (2020). We train for 30 epochs on CIFAR Krizhevsky & Hinton (2009) and 15 epochs for SVHN Netzer et al. (2011) following Andriushchenko & Flammarion (2020). When we perform PGD-AT we use 10 steps and a step size $\alpha = 2/255$; FGSM uses a step size of $\alpha = \epsilon$. Regularization parameters for GradAlign Andriushchenko & Flammarion (2020) and N-FGSM de Jorge et al. (2022) will vary and are stated when relevant in the paper. The architecture employed is a PreactResNet18 He et al. (2016). Robust accuracy is evaluated by attacking the trained models with PGD-50-10, i.e. PGD with 50 iterations and 10 restarts. In this case we also employ a step size $\alpha = 2/255$ as in Wong et al. (2020). All accuracies are averaged after training and evaluating with 3 random seeds.

The curvature computation is performed following the procedure described in Moosavi-Dezfooli et al. (2019). As they propose, we use finite differences to estimate the directional second derivative of the loss with respect to the input, *i.e.,*

$$\boldsymbol{w}^\top \nabla_{\boldsymbol{x}}^2 \mathcal{L}(f_{\boldsymbol{\theta}}(\boldsymbol{x}), y) \approx \frac{\nabla_{\boldsymbol{x}} \mathcal{L}(f_{\boldsymbol{\theta}}(\boldsymbol{x} + t\boldsymbol{w}), y) - \nabla_{\boldsymbol{x}} \mathcal{L}(f_{\boldsymbol{\theta}}(\boldsymbol{x} - t\boldsymbol{w}), y)}{2t},$$

with $t > 0$ and use the Lanczos algorithm to perform a partial eigendecomposition of the Hessian without the need to compute the full matrix. In particular, we pick $t = 0.1$.

All our experiments were performed using a cluster equipped with GPUs of various architectures. The estimated compute budget required to produce all results in this work is around $2,000$ GPU hours (in terms of NVIDIA V100 GPUs).

## D  INDUCING CATASTROPHIC OVERFITTING WITH OTHER SETTINGS

In Section 3 we have shown that CO can be induced with data interventions for CIFAR-10 and $\ell_\infty$ perturbations. Here we present similar results when using other datasets (i.e. CIFAR-100 and SVHN) and other types of perturbations (i.e. $\ell_2$ attacks). Moreover, we also report results when the injected features $\boldsymbol{v}(y)$ follow random directions (as opposed to low-frequency DCT components). Overall, we find similar findings to those reported the main text.

### D.1  OTHER DATASETS

Similarly to Section 3 we modify the SVHN, CIFAR-100 and high resolution Imagenet-100 and TinyImagenet datasets to inject highly discriminative features $\boldsymbol{v}(y)$. Since SVHN also has 10 classes, we use the exact same settings as in CIFAR-10 and we train and evaluate with $\epsilon = 4$ where training on the original data does not lead to CO (recall $\beta = 0$ corresponds to the unmodified dataset). On the other hand, for CIFAR-100 and Imagenet-100 we select $\boldsymbol{v}(y)$ to be the 100 DCT components with lowest frequency and we present results with $\epsilon = 5$ and $\epsilon = 4$ respectively. Similarly, for TinyImagenet (which has 200 classes) we use the first 200 DCT components and present results with $\epsilon = 6$. Moreover, for ImageNet-100 we evaluate robustness with AutoAttack Croce & Hein (2020). Regarding the training settings, for CIFAR10/100 and SVHN datasets we use the same settings as Andriushchenko & Flammarion (2020), for ImageNet-100 we follow Kireev et al. (2022) and for TinyImageNet Li et al. (2020).

In all datasets we can observe similar trends as with CIFAR-10: For small values of $\beta$ the injected features are not very discriminative due to their interaction with the dataset images and the model largely ignores them. As we increase $\beta$, there is a range in which they become more discriminative but not yet robust and we observe CO. Finally for large values of $\beta$ the injected features become robust and the models can achieve very good performance focusing only on those.

### D.2  OTHER NORMS

Catastrophic overfitting has been mainly studied for $\ell_\infty$ perturbations and thus we presented experiments with $\ell_\infty$ attacks following related work. However, in this section we also present results where

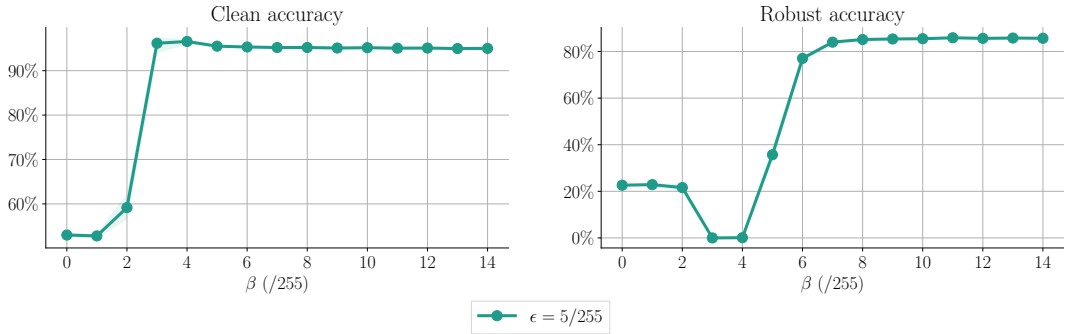

Figure 8: Clean and robust performance after FGSM-AT on injected datasets $\widetilde{\mathcal{D}}_\beta$ constructed from CIFAR-100. As FGSM-AT already suffers CO on CIFAR-100 at $\epsilon = {}^6/255$ we use $\epsilon = {}^5/255$ in this experiment where FGSM-AT does not suffer from CO as seen for $\beta = 0$. In this setting, we observe CO happening when $\beta$ is slightly smaller than $\epsilon$. Results are averaged over 3 seeds and shaded areas report minimum and maximum values.

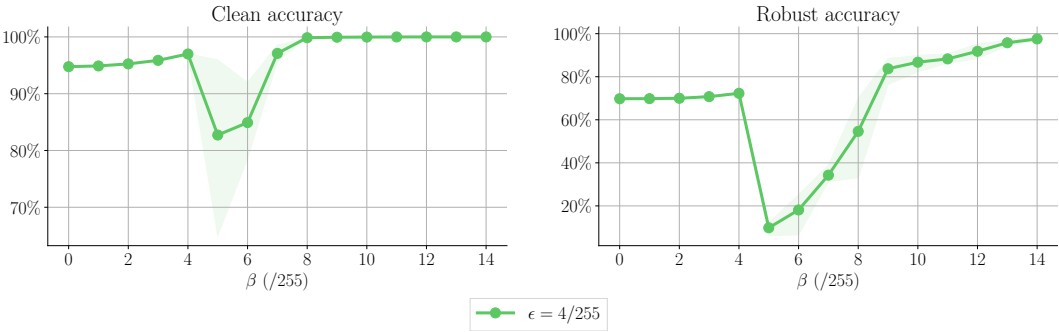

Figure 9: Clean and robust performance after FGSM-AT on injected datasets $\widetilde{\mathcal{D}}_\beta$ constructed from SVHN. As FGSM-AT already suffers CO on SVHN at $\epsilon = {}^6/255$ we use $\epsilon = {}^4/255$ in this experiment where FGSM-AT does not suffer from CO as seen for $\beta = 0$. In this setting, we observe CO happening when $\beta \approx \epsilon$. Results are averaged over 3 seeds and shaded areas report minimum and maximum values.

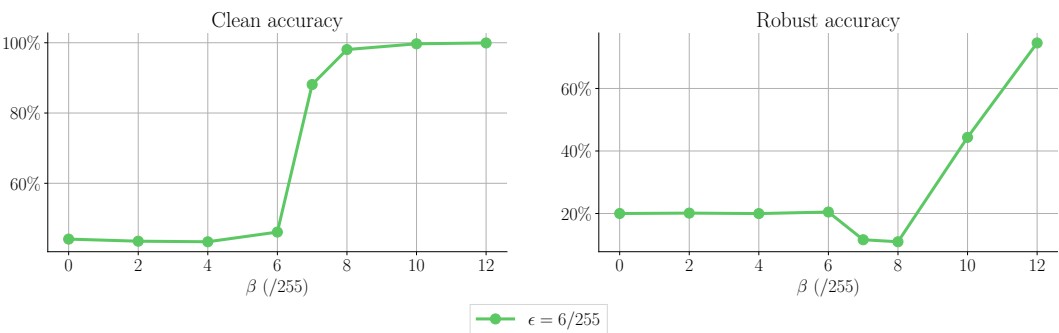

Figure 10: Clean and robust performance after FGSM-AT on injected datasets $\widetilde{\mathcal{D}}_\beta$ constructed from TinyImageNet. We use $\epsilon = {}^6/255$ in this experiment where FGSM-AT does not suffer from CO as seen for $\beta = 0$. In this setting, we observe CO happening for $\beta \in [{}^7//255, {}^8//255]$.

we induce CO with $\ell_2$ perturbation which are also widely used in adversarial robustness. In Fig. 11 we show the clean (left) and robust (right) accuracy after FGM-AT[5] on our injected dataset from CIFAR-10 ($\widetilde{\mathcal{D}}_\beta$). Similarly to our results with $\ell_\infty$ attacks, we also observe CO as the injected features

---

[5]FGM is the $\ell_2$ version of FGSM where we do not take the sign of the gradient.

Table 2: Test performance of FGSM-AT trained on different injected versions of ImageNet-100 with $\epsilon = 4/255$. We observe that for $\beta > \epsilon$, FGSM-AT clearly suffers from CO.

|  | Clean accuracy | FGSM accuracy | Robust accuracy |
|---|---|---|---|
| $\beta = 0/255$ | 71.0% | 49.1% | 45.6% |
| $\beta = 2/255$ | 70.9% | 51.1% | 45.1% |
| $\beta = 4/255$ | 74.2% | 52.6% | 42.1% |
| $\beta = 5/255$ | 100% | 99.8% | 0.0% |
| $\beta = 6/255$ | 100% | 98.0% | 1.9% |

become more discriminative (increased $\beta$). It is worth mentioning that the $\ell_2$ norm we use ($\epsilon = 1.5$) is noticeably larger than typically used in the literature, however, it would roughly match the magnitude of an $\ell_\infty$ perturbation with $\epsilon = 7/255$. Interestingly, we did not observe CO for this range of $\beta$ with $\epsilon = 1$.

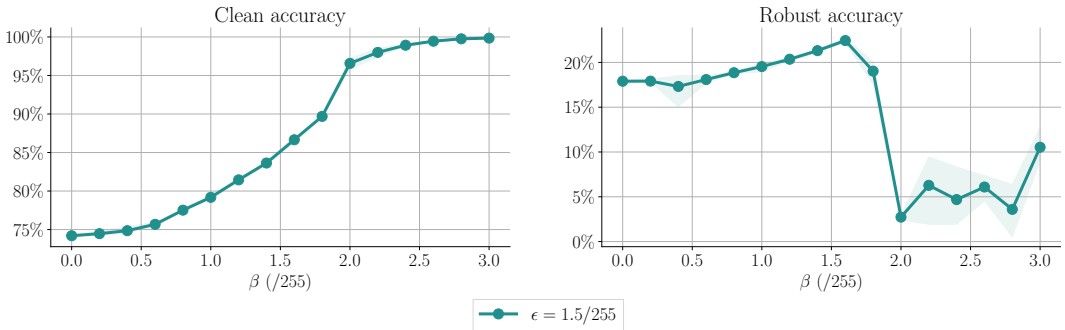

Figure 11: Clean and $\ell_2$ robust performance after FGSM-AT on injected datasets $\widetilde{\mathcal{D}}_\beta$ constructed from CIFAR-10. FGM-AT suffers CO on CIFAR-10 around $\epsilon = 2$, so we use $\epsilon = 1.5$ in this experiment where FGM-AT does not suffer from CO as seen for $\beta = 0$. In this setting, we observe CO happening when $\beta \approx \epsilon$. Results are averaged over 3 seeds and shaded areas report minimum and maximum values.

### D.3   OTHER INJECTED FEATURES

We selected the injected features for our injected dataset from the low frequency components of the DCT to ensure an interaction with the features present on natural images Ahmed et al. (1974). However, this does not mean that other types of features could not induce CO. In order to understand how unique was our choice of features we also created another family of injected datasets but this time using a set of 10 randomly generated vectors as features. As in the main text, we take the sign of each random vector to ensure they take values in $\{-1, +1\}$ and assign one vector per class. In Fig. 12 we observe that using random vectors as injected features we can also induce CO. Note that since our results are averaged over 3 random seeds, each seed uses a different set of random vectors.

### D.4   OTHER ARCHITECTURES

In all our previous experiments we trainde a PreActResNet18 (He et al., 2016) as it is the standard architecture used in the literature. However, our observations our also robust to the choice of architecture. As we can see in Fig. 13, we can also induce CO when training a WideResNet28x10 Zagoruyko & Komodakis (2016) on an injected version of CIFAR-10.

## E   LEARNED FEATURES AT DIFFERENT $\beta$

In Section 3 we discussed how, based on the strength of the injected features $\beta$, our injected datasets seem to have 3 distinct regimes: (i) When $\beta$ is small we argued that the network would not use the

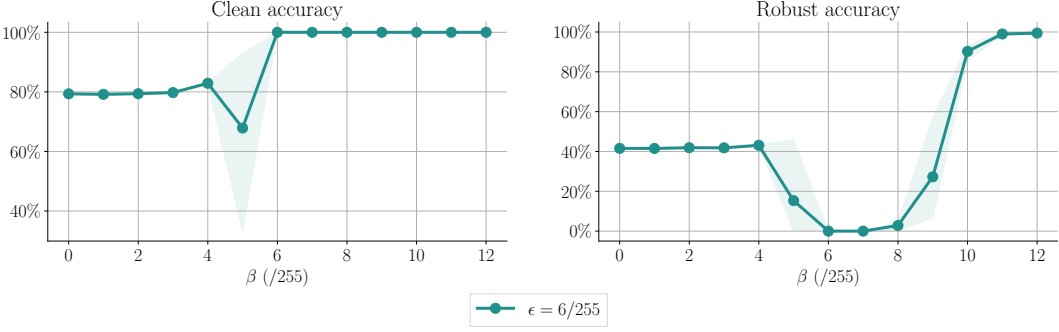

Figure 12: Clean and robust performance after FGSM-AT on injected datasets $\widetilde{\mathcal{D}}_\beta$ constructed from CIFAR-10 using random signals in $\mathcal{V}$. We perform this experiments at $\epsilon = {}^6/_{255}$ where we saw that injected the dataset with the DCT basis vectors did induce CO. In the random $\mathcal{V}$ setting, we observe the same behaviour, with CO happening when $\beta \approx \epsilon$. Results are averaged over 3 seeds and shaded areas report minimum and maximum values.

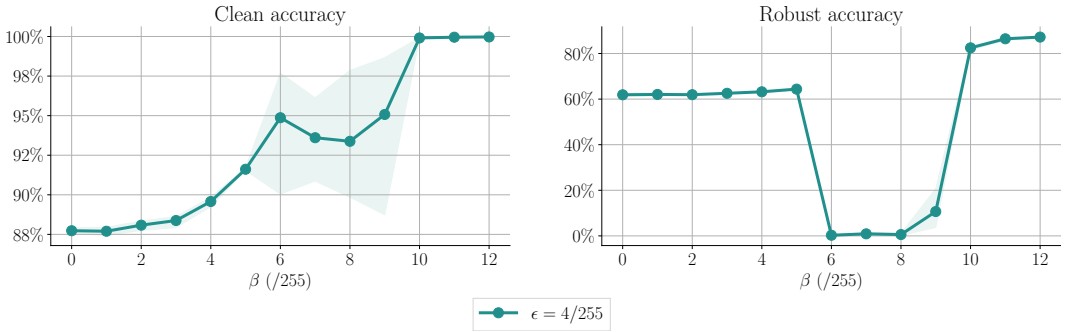

Figure 13: Clean and robust performance after FGSM-AT on injected datasets $\widetilde{\mathcal{D}}_\beta$ constructed from CIFAR-10 when training a WideResNet28x10. We perform this experiments at $\epsilon = {}^4/_{255}$. Results are averaged over 3 seeds and shaded areas report minimum and maximum values.

injected features as these would not be very helpful. (ii) When $\beta$ would have a very large value then the network would only look at these features since they would be easy-to-learn and provide enough margin to classify robustly. (iii) Finally, there was a middle range of $\beta$ usually when $\beta \sim \epsilon$ where the injected features would be strongly discriminative but not enough to provide robustness on their own. This regime is where we observe CO.

In this section we present an extension of Fig. 3 where we take FGSM trained models on the injected datasets ($\widetilde{\mathcal{D}}_\beta$) and evaluate them on three test sets: (i) The injected test set ($\widetilde{\mathcal{D}}_\beta$) with the same features as the training set. (ii) The original dataset ($\mathcal{D}$) where the images are unmodified. (iii) The shuffled dataset ($\widetilde{\mathcal{D}}_{\pi(\beta)}$) where the injected features are permuted. That is, the set of injected features is the same but the class assignments are shuffled. Therefore, the injected features will provide conflicting information with the features present on the original image.

In Fig. 14 we show the performance on the aforementioned datasets for three different values of $\beta$. For $\beta = {}^2/_{255}$ we are in regime (i) : we observe that the tree datasets have the same performance, i.e. the information of the injected features does not seem to alter the result. Therefore, we can conclude the network is mainly using the features from the original dataset $\mathcal{D}$. When $\beta = {}^{20}/_{255}$ we are in regime (ii) : the clean and robust performance of the network is almost perfect on the injected test set $\widetilde{\mathcal{D}}_\beta$ while it is close to $0\%$ (note this is worse than random classifier) for the shuffled dataset. So when the injected and original features present conflicting information the network aligns with the injected features. Moreover, the performance on the original dataset is also very low. Therefore, the network is mainly using the injected features. Lastly, $\beta = {}^8/_{255}$ corresponds to regime (iii) : as discussed in Section 4.1, in this regime the network initially learns to combine information from both the original

and injected features. However, after CO, the network seems to focus only on the injected features and discards the information from the original features.

Figure 14: Clean (**top**) and robust (**bottom**) accuracy of FGSM-AT on $\widetilde{\mathcal{D}}_\beta$ at different $\beta$ values on 3 different test sets: (i) the original CIFAR-10 ($\mathcal{D}$), (ii) the dataset with injected features $\widetilde{\mathcal{D}}_\beta$ and (iii) the dataset with shuffled injected features $\widetilde{\mathcal{D}}_{\pi(\beta)}$. All training runs use $\epsilon = 6/255$. **Left**: $\beta = 2/255$ **Center**: $\beta = 8/255$ **Right**: $\beta = 20/255$.

## F    ANALYSIS OF CURVATURE IN DIFFERENT SETTINGS

In Fig. 4 (left) we track the curvature of the loss surface while training on different injected datasets with either PGD-AT or FGSM-AT. We observe that (i) Curvature rapidly increases both for PGD-AT and FGSM-AT during the initial epochs of training. (ii) In those runs that presented CO, the curvature explodes around the $8^{th}$ epoch along with the training accuracy. (iii) When training with the dataset with orthogonally injected features ($\widetilde{\mathcal{D}}_\beta^\perp$) the curvature does not increase. This is aligned with our proposed mechanisms to induce CO whereby the network increases the curvature in order to combine different features to learn better representations. In this section we extend this analysis to the original CIFAR-10 dataset (as opposed to our injected datasets) and to different values of feature strength $\beta$ on the injected dataset ($\widetilde{\mathcal{D}}_\beta$). For details on how we estimate the curvature refer to Appendix C.

In Fig. 15 we show the curvature when training on the original CIFAR-10 dataset with $\epsilon = 8/255$ (where CO happens for FGSM-AT). Similarly to our observations on the injected datasets, the curvature during FGSM-AT explodes along with the training accuracy while for PGD-AT the curvature increases at a very similar rate than FGSM-AT during the first epochs and later stabilizes. This indicates that our described mechanisms may as well apply to induce CO on natural image datasets.

On the other hand, Fig. 16 presents the curvature for different values of feature strength $\beta$ on the injected dataset ($\widetilde{\mathcal{D}}_\beta$). We show three different values of $\beta$ representative of the three regimes discussed in Appendix E. Recall that when $\beta$ is small ($\beta = 2/255$) we observe that the model seems to focus only on CIFAR-10 features. Thus, we observe a curvature increase aligned with (CIFAR-10) feature combination. However, since for the chosen robustness radii $\epsilon = 6/255$ there is no CO, we observe that the curvature increase remains stable. When $\beta$ is quite large ($\beta = 20/255$) then the model largely ignores CIFAR-10 information and focuses on the easy-to-learn injected features. Since these features are already robust, there is no need to combine them and the curvature does not need to increase. In the middle range when CO happens ($\beta = 8/255$) we observe again the initial curvature increase and then curvature explosion.

## G    ADVERSARIAL PERTURBATIONS BEFORE AND AFTER CO

**Qualitative analysis**    In order to further understand the change in behaviour after CO we presented visualizations of the FGSM perturbations before and after CO in Fig. 2. We observed that while prior

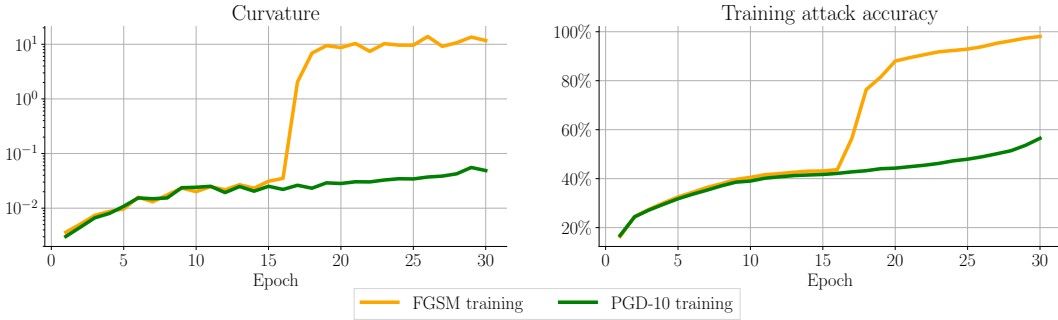

Figure 15: Evolution of curvature and training attack accuracy of FGSM-AT and PGD-AT trained on the original CIFAR-10 with $\epsilon = 8/255$. When CO happens the curvature explodes.

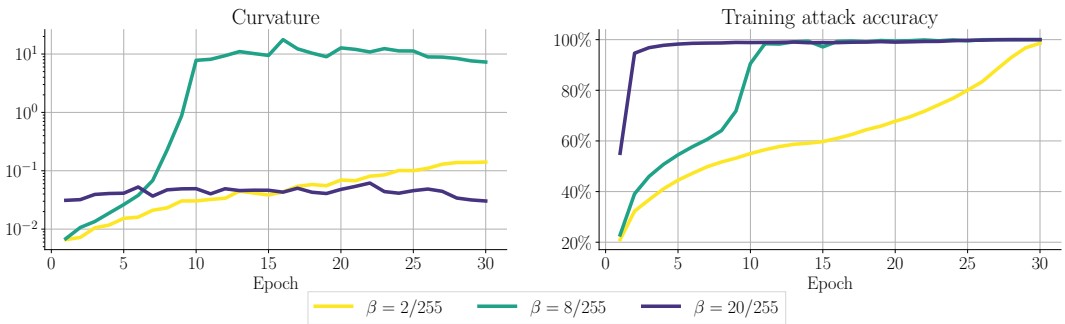

Figure 16: Evolution of curvature and training attack accuracy of FGSM-AT and PGD-AT trained on $\widetilde{\mathcal{D}}_\beta$ at different $\beta$ and for $\epsilon = 6/255$. Only when CO happens (for $\beta = 8/255$) the curvature explodes. For the other two interventions the curvature does not increase so much. We argue this is because the network does not need to disentangle $\mathcal{D}$ from $\mathcal{V}$, as it ignores either one of them.

to CO, the injected feature components $v(y)$ were clearly identifiable, after CO the perturbations do not seem to point in those directions although the network is strongly relying on them to classify. In Fig. 17 and Fig. 18 we show further visualizations of the perturbations obtained both with FGSM or PGD attacks on networks trained with either PGD-AT or FGSM-AT respectively.

We observe that when training with PGD-AT, i.e. the training does not suffer from CO, both PGD and FGSM attacks produce qualitatively similar results. In particular, all attacks seem to target the injected features with some noise due to the interaction with the features from CIFAR-10. For FGSM-AT, we observe that at the initial epochs (prior to CO) the pertubations are similar to those of PGD-AT, however, after CO perturbations change dramatically both for FGSM and PGD attacks. This aligns with the fact that the loss landscape of the network has dramatically changed, becoming strongly non-linear. This change yields single-step FGSM ineffective, however, the network remains vulnerable and multi-step attacks such as PGD are still able to find adversarial examples, which in this case do not point in the direction of discriminative features Jetley et al. (2018); Ilyas et al. (2019).

**Quantitative analysis** Finally, to quantify the radical change of direction of the adversarial perturbations after CO, we compute the evolution of the average alignment (*i.e.,* cosine angle) between the FGSM perturbations $\delta$ and the injected features, such that if point $x$ is associated with class $y$ we compute $\frac{\langle \delta, v(y) \rangle}{\|v(y)\|_2 \|\delta\|_2}$. Figure 19 (left) shows the results of this evaluation, where we can see that before CO there is a non-trivial alignment between the FGSM perturbations and their corresponding injected features, that after CO quickly converges to the same alignment as the one between two random vectors.

To complement this view, we also perform an analysis of the frequency spectrum of the FGSM perturbations. In Fig. 19 (right), we plot the average magnitude of the DCT transform of the FGSM perturbations computed on the test set of an intervened version of CIFAR-10 during training. As we can see, prior to CO, most of the energy of the perturbations is concentrated around the low frequencies (remember that the injected features are low frequency), but after CO happens, around epoch 8, the

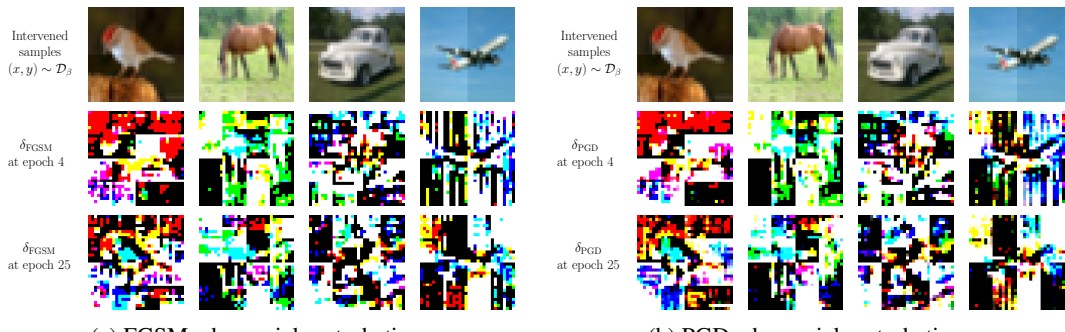

(a) FGSM adversarial perturbations        (b) PGD adversarial perturbations

Figure 17: Different samples of the injected dataset $\widetilde{\mathcal{D}}_\beta$, and adversarial perturbations at epoch 4 and 22 of PGD-AT on $\widetilde{\mathcal{D}}_\beta$ at $\epsilon = {}^6/_{255}$ and $\beta = {}^8/_{255}$ (where FGSM-AT suffers CO). The adversarial perturbations remain qualitatively similar throughout training and align significantly with $\mathcal{V}$.

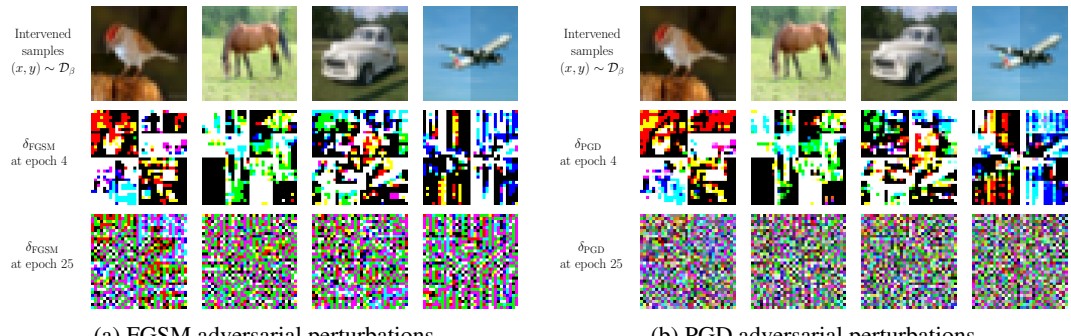

(a) FGSM adversarial perturbations        (b) PGD adversarial perturbations

Figure 18: Different samples of the injected dataset $\widetilde{\mathcal{D}}_\beta$, and adversarial perturbations at epoch 4 (before CO) and 22 (after CO) of FGSM-AT on $\widetilde{\mathcal{D}}_\beta$ at $\epsilon = {}^6/_{255}$ and $\beta = {}^8/_{255}$ (where FGSM-AT suffers CO). The adversarial perturbations change completely before and after CO. Prior to CO, they align significantly with $\mathcal{V}$, but after CO they point to meaningless directions.

energy of the perturbations quickly gets concentrated towards higher frequencies. These two plots corroborate, quantitatively, our previous observations, that before CO, FGSM perturbations are pointing towards meaningful predictive features, while after CO, although we know the network still uses the injected features (see Fig. 3) the FGSM perturbations suddenly point in a different direction.

## H  FURTHER RESULTS WITH N-FGSM, GRADALIGN AND PGD

In Section 5 we studied different SOTA methods that have been shown to prevent CO. Interestingly, we observed that in order to avoid CO on the injected dataset a stronger level of regularization is needed. Thus, indicating that the intervention is strongly favouring the mechanisms that lead to CO. For completeness, in Fig. 20 we also present results of the clean accuracy (again with the robust accuracy). As expected, for those runs in which we observe CO, clean accuracy quickly saturates. Note that for stronger levels of regularization the clean accuracy is lower. An ablation of the regularization strength might help improve results further, however the purpose of this analysis is not to improve the performance on the injected dataset but rather to show it is indeed possible to prevent CO with the same methods that work for unmodified datasets.

## I  FURTHER DETAILS OF LOW-PASS EXPERIMENT

We expand here over the results in Section 5 and provide further details on the experimental settings of Table 1. Specifically, we replicate the same experiment, *i.e.,* training a low-pass version of CIFAR-10

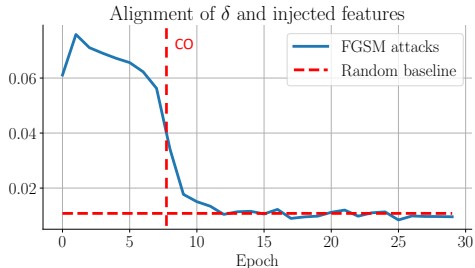 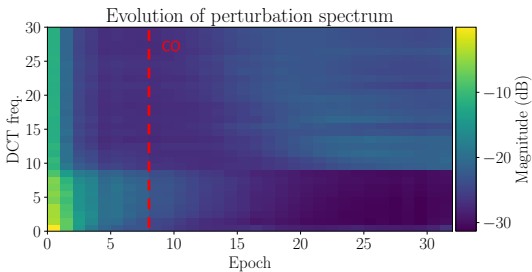

Figure 19: Quantitative analysis of the directionality of the FGSM perturbations in the test set during FGSM-AT before and after CO when training on the injected CIFAR-10 with $\beta = {}^8/_{255}$ and $\epsilon = {}^6/_{255}$. (**Left**) Evolution of alignment of FGSM perturbations with their corresponding injected features during training. The red dotted line shows the expected value of alignment between two random vectors of the same dimensionality as CIFAR-10 images. (**Right**) Evolution of the average magnitude of the DCT spectrum of the same FGSM perturbations during training. The plot shows only the diagonal components of the DCT at every epoch.

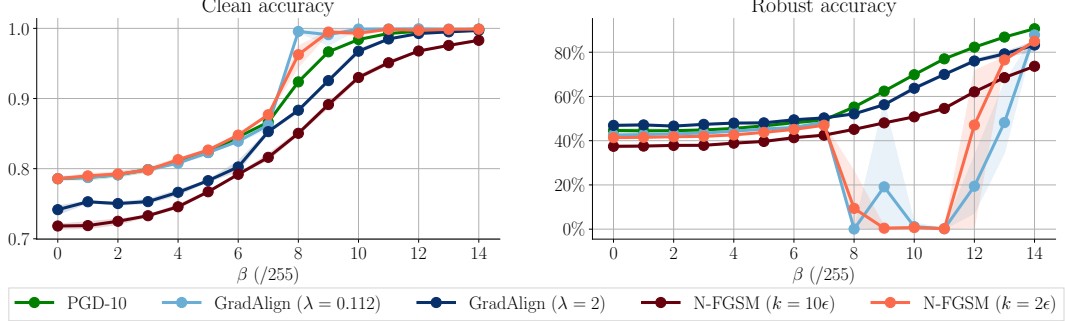

Figure 20: Clean (**left**) and robust (**right**) accuracy after AT with PGD-10, GradAlign and N-FGSM on $\widetilde{\mathcal{D}}_\beta$ at $\epsilon = {}^6/_{255}$. Results averaged over three random seeds and shaded areas report minimum and maximum values.

using FGSM-AT at different filtering bandwidths. As indicated in Section 5 we use the low-pass filter introduced in Ortiz-Jimenez et al. (2020b) which only retains the frequency components in the upper left quadrant of the DCT transform of an image. That is, a low-pass filter of bandwidth $W$ would retain the $W \times W$ upper quadrant of DCT coefficients of all images, setting the rest of the coefficients to 0.

Figure 21 shows the robust accuracy obtained by FGSM-AT on CIFAR-10 versions that have been pre-filtered using such a low-pass filter. Interestingly, while training on the unfiltered images does induce CO on FGSM-AT, just removing a few high-frequency components is enough to prevent CO $\epsilon = 8255$. However, as described before, it seems that at $\epsilon = {}^{16}/_{255}$ no frequency transformation can avoid CO. Clearly, this transformation cannot be used as technique to prevent CO, but it highlights once again that the structure of the data plays a significant role in inducing CO.

**Relation with anti-aliasing pooling layers**    As mentioned in Section 5, our proposed low-passing technique is very similar in spirit to works which propose using anti-aliasing low-pass filters at all pooling layers (Grabinski et al., 2022; Zhang, 2019). Indeed, as shown by Ortiz-Jimenez et al. (2020b), CIFAR10 contains a significant amount of non-robust features on the high-frequency end of the spectrum due to aliasing produced in their down-sampling process. In this regard, it is no surprise that methods like the one proposed in Grabinski et al. (2022) can prevent CO at $\epsilon = {}^8/_{255}$ using the same training protocol as in our work (robust accuracy is 45.9%). Interestingly, though, repeating the experiments in Grabinski et al. (2022) work using $\epsilon = {}^{16}/_{255}$ does lead to CO (robust accuracy is 0.0%) in Section 5. This result was not reported in the original paper, but we see it as a corroboration of our observations. Indeed, features play a role in CO, but the problematic features

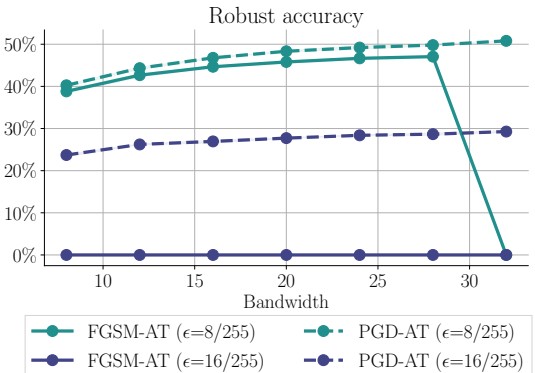

Figure 21: Robust accuracy of FGSM-AT and PGD-AT on different low-passed versions of CIFAR-10 using the DCT-based low pass filter introduced in Ortiz-Jimenez et al. (2020b). Bandwidth $= 32$ corresponds to the original CIFAR-10, while smaller bandwidths remove more and more high-frequency components. At $\epsilon = 8/255$ just removing a few high-frequency components is enough to prevent CO, while at $\epsilon = 16/255$ no frequency transformation avoids CO.

do not always come from excessively high-frequencies or aliasing. However, we still consider that preventing aliasing in the downsampling layers is a promising avenue for future work in adversarial robustness.

