# OpenReview forum: "Catastrophic overfitting is a bug but it is caused by features"
_ICLR.cc/2023/Conference — Submitted to ICLR 2023_

### Official Review · Reviewer_GG1w · 2022-10-24

**Confidence:** 4
**Clarity, Quality, Novelty And Reproducibility:** W1. Over-claim of the contribution in…
**Correctness:** 3
**Technical Novelty And Significance:** 3
**Empirical Novelty And Significance:** 3
**Recommendation:** 6

**Strength And Weaknesses:**

S1. The experiments are well designed and convincing.
S2. The theory for onset of CO is both empirically and theoretically justified.
S3. The related work is well cited and discussed.


**Summary Of The Paper:**

The authors tried to explain the cause of catastrophic overfitting (CO) during FGSM adversarial training. The authors first showed that CO can be deliberately induced by dataset intervention. Then, using this intervened dataset as a starting point, the authors conducted three well-designed experiments to study the chain of mechanisms behind CO. They conclude that in FGSM-AT:
1. The network attempts to learn both non-robust and robust features.
2. The network increases its non-linearity for better learning these two types of features.
3. The explosion of non-linearity breaks FGSM, which provides a shortcut for networks to only learn from non-robust features, which eventually leads to CO.



**Summary Of The Review:**

Overall it’s a solid and well written paper. The way the authors design experiments is novel. Both empirical results and theoretical justifications are provided. And they successfully fill the knowledge gap on the onset of CO. The only weakness is that they seem to over-claim their contribution in dataset construction. The authors stated that it’s hard to separate robust features from non-robust features in real datasets, so they used their way of dataset construction. But (Ilyas et al., 2019) seems to have successfully disentangled these two features from neural networks’ perspective. In (Ilyas et al., 2019), a method is provided to extract non-robust features and robust features. So why don't the authors use their methodology?

---

> ### Author Response · Authors · 2022-11-10
> **Response to reviewer GG1w**
>
> We would like to thank reviewer GG1w for their comments. We appreciate that they found our experiments to be well designed and convincing, our explanation for the onset of CO both empirically and theoretically justified, and that our related work is well referenced and discussed. We would like to point the reviewer to our [general message to all reviewers](https://openreview.net/forum?id=n0okuXMlI7V&noteId=e1vUYSktrZY). In what follows we address the particular concern of this reviewer.
>
> > Why don't the authors use Ilyas et al., 2019 to extract the non-robust features of the dataset?
>
> Regarding the main concern, we want to emphasise that this is indeed an excellent idea that we hope other works can explore further. However, we would like to highlight that **Ilyas et al.’s approach is still synthetically intervening in the data**: It adds an adversarial perturbation to each datapoint and flips the label to the adversarial class. If we repeat this process in our work, the concerns about our experiments being performed on artificial datasets would not be alleviated by following this approach.
>
> In any case, we did not directly apply this methodology in our study because it cannot directly be used to test our observations. **A fundamental tool required to observe our discoveries is to be able to identify and play with the strength of the easy features**. However, Ilyas et al. method only allows to toggle on/off the robust features, regardless of their simplicity. In this sense, our intervention allows us to modify the robustness of the injected features directly (through $\beta$), without affecting their easy-to-learn nature (in the clean sense). Furthermore, we believe that our additive features are more interpretable, and easy to visualise than input-dependent adversarial perturbations. This is key for instance to be able to visually identify the injected features in the FGSM attacks in Figure 2.
>
> Overall, **we believe that performing our experiments using Ilyas et al.’s approach would have made disentangling all these characteristics more challenging and our results less intuitive**. Therefore, we humbly believe that for our specific objectives, our data intervention was a more sensible approach, but, we sincerely thank the reviewer for raising this point. This is a promising avenue for future work and we have clearly mentioned it in our new conclusions.
>
> We hope we have addressed the reviewer's concerns so we politely ask the reviewer to consider increasing their score.

---

### Official Review · Reviewer_2rv5 · 2022-11-02

**Confidence:** 5
**Clarity, Quality, Novelty And Reproducibility:** Clear lack of novelty -> see above
**Correctness:** 2
**Technical Novelty And Significance:** 2
**Empirical Novelty And Significance:** 2
**Recommendation:** 5

**Strength And Weaknesses:**

Strengths:
* the paper addresses an important and interesting question
* the paper introduces the OT problem well and gives a quite good overview of the related work (missing only a view important works [1-3])

[3] Grabinski et. all: FrequencyLowCut Pooling--Plug & Play against Catastrophic Overfitting, ECCV 2022

Weaknesses:

1) the title is overclaiming: overfitting is NOT a bug and the paper fails to show that OT is indeed caused by features
2) novelty: the paper fails to show new insights: OT is overfitting - this is in line with all related work, all known counter measures (augmentation, regularization, input smoothing ...) and the presented experiments only confirm this
3) experimental setup is not suited to answer the questions defined at the beginning of the paper. Adding easy to learn patterns to the data, leading 100% clean ACC is definitely provoking ofervitting. The described case of beta=eps does not allow strong conclusions other than OT is overfitting. Additional results on the nonlinearity and non-smoothness of the optimization space only further confirm this (and have been shown before)


**Summary Of The Paper:**

the paper investigates possible causes of catastrophic overfitting (CO), a well known state in adversarial training (AT) of robust model which leads to the collapse of model robustness during later stages of AT. The authors propose to augment the data with easy to classify DCT patterns during analisis, in order to gain insights on the properties of the feature space leading to OT.

**Summary Of The Review:**

the initial question of how the feature space representation affects the CO is indeed very interesting and requires further investigation. However, I think that presented approach is not suitable to achieve this goal because the chosen augmentation of training data will trigger clean overfitting and thus prevents an unbiased investigation.

I would suggest that the authors investigate the features at CO without external manipulation. [1+2] provided a large DB of models which could be used to conduct these studies and have already shown that robust features are only different in specific layers of a network...

[1] Gavrikov et. all Adversarial Robustness through the Lens of Convolutional Filters, CVPR-W 2022

[2] Gavrikov et. all. CNN Filter DB: An Empirical Investigation of Trained Convolutional Filters, CVPR 2022

---

> ### Author Response · Authors · 2022-11-10
> **Response to reviewer 2rv5 (Part 1 of 2)**
>
> We thank reviewer 2rv5 for their comments. We appreciate that they find our work to be addressing a very important and interesting question. First, we would like to point the reviewer to the [general message](https://openreview.net/forum?id=n0okuXMlI7V&noteId=e1vUYSktrZY) we posted where we discuss one of the referenced works [3] in their review.
> In what follows we expand more on the reviewer’s specific comments:
>
> > The title is overclaiming.
>
> Our title is a reference to Ilyas et. al. [4] (Adversarial examples are not bugs, they are features) as our study is partially inspired by their work. When we say that CO is a bug, we are referring to the clear consensus of the community that **CO is an undesirable failure mode of FGSM adversarial training which is not intended by design**. Regarding the fact that it is caused by features, in our experiments, we do provide thorough empirical evidence that the **injection of features in the data can cause CO in situations where it does not occur otherwise**. This leads to a novel perspective of CO. We discuss later the suitability of these experiments.
>
> > Experimental setup.
>
> Despite its name, **Catastrophic Overfitting is not simply “overfitting” in the classical sense of the term**. The definition of overfitting requires there to be a large gap between train and test accuracy. If a model has 100% train accuracy AND 100% test accuracy, then it is not classical overfitting. See Figure 3 in our work: For FGSM, the top figure shows that the model has 100% clean test accuracy (thus has not overfitted), while in the bottom of Figure 3 we show that the model has 0% robust accuracy.
>
> Moreover, CO is not robust overfitting either. As noted in [5] (appendix A.4.), even when avoiding CO, FGSM-AT is still susceptible to robust overfitting.
>
> We want to highlight that Wong et. al. [6] initially coined the term Catastrophic Overfitting because they thought this phenomenon was due to the models “overfitting”  to the low variability in FGSM perturbations. However, please note that Andriuschenko and Flammarion [7] later showed this is not true, as they showed that PGD AT with attacks projected to the corners of the $\ell_\infty$ ball did not lead to CO. Still the term Catastrophic Overfitting remained, despite referring to a very different effect to classical overfitting.
> As it is well known by the community, the absence of adversarial robustness is linked to the preference of the networks to learn non-robust features [4]. In this work we build on this intuition to design our data intervention by injecting easy-to-learn features. Nonetheless, it is crucial to note that the biases of the network for some features over others are not due to overfitting since our injected features are present both during training and test time. In this regard, we strongly believe that our intervention is a valid strategy to study the effect that easy-to-learn features have on FGSM training in a controlled scenario, where we can vary their robustness. In fact, **our data intervention is not leveraging any bias in the network aside from the one we want to study (i.e., preference for easy-to-learn features)**.
>
> > Novelty.
>
> Having established that CO is not mere “overfitting”, **we do think that our insights are indeed novel since no prior work had established the cause for the large increase in non-linearity associated with CO** (which we agree with the reviewer, was already known). In our experiments, we study the phenomena of CO while knowing which intervention caused it (since we study $\epsilon$ radii where CO does not happen without intervention). This allows us to focus the lens of study on the right direction linking the different known observations about CO in a novel view as highlighted by other reviewers.
>
> *Our answer continues below.*

---

> > ### Author Response · Authors · 2022-11-10
> > **Response to reviewer 2rv5 (Part 2 of 2)**
> >
> >
> > > Missing references to (Gabrikov & Keuper, 2022a, Gabrikov & Keuper, 2022b).
> >
> > Regarding the two references to the work of Gabrikov & Keuper [1, 2] and possible experiments on the features at intermediate layers, we would like to thank the reviewer for the suggestion. However, we do not think that these works have such a close relation to catastrophic overfitting that justifies the reviewer to point them out as important related work (**these works do not discuss “catastrophic overfitting", in fact they do not even mention it anywhere on their text**). As such, although interesting, these are only tangentially related to our work as they deal with the interpretability/explainability of robust models. As the reviewer would agree, there are hundreds of papers dealing with that particular topic, and citing them all would be impossible (this recent survey [8] on that field contains more than 200 references). In this regard, **making a specific reference to these two particular works would be certainly unfair to the rest of the community**. However, if the reviewer feels very strongly about it, we would be open to discuss this field in our related work.
> >
> > Regarding the suggested experiments, we agree with the reviewer that studying the feature maps at intermediate layers of the network in the context of CO would indeed be interesting. However, please note that **this is a different direction from our study which investigates the role of input features, not feature maps**. Moreover, note that studying the feature maps with or without CO would again lead to an observational stance, however, our intervention allows us to study the change (injecting features) that led to CO.
> > In any case, we thank 2rv5 for their suggestion. We will consider it as a direction for future work. We also hope other people in the community can expand on these ideas if our work is published.
> >
> > [1] Paul Gavrikov and Janis Keuper, CNN Filter DB: An Empirical Investigation of Trained Convolutional Filters, CVPR 2022
> >
> > [2] Paul Gavrikov and Janis Keuper, Adversarial Robustness through the Lens of Convolutional Filters, CVPR-W 2022
> >
> > [3] Julia Grabinski, Steffen Jung, Janis Keuper, and Margret Keuper. FrequencyLowCut Pooling - Plug & Play against Catastrophic Overfitting. ECCV, (October) 2022.
> >
> > [4] Andrew Ilyas, Shibani Santurkar, Dimitris Tsipras, Logan Engstrom, Brandon Tran, and Aleksander Madry. Adversarial examples are not bugs, they are features. Neural Information Processing Systems (NeurIPS), 2019.
> >
> > [5] Leslie Rice, Eric Wong, and Zico Kolter. Overfitting in adversarially robust deep learning. In International Conference on Machine Learning (ICML), 2020
> >
> > [6] Eric Wong, Leslie Rice, and J. Zico Kolter. Fast is better than free: Revisiting adversarial training. In International Conference on Learning Representations (ICLR), 2020
> >
> > [7] Andriuschenko and Flammarion, Understanding and improving fast adversarial training, NeurIPS 2020
> >
> > [8] Guillermo Ortiz-Jiménez, Apostolos Modas, Seyed-Mohsen Moosavi-Dezfooli, and Pascal Frossard. Optimism in the face of adversity: Understanding and improving deep learning through adversarial robustness. Proceedings of the IEEE, 2021.

---

> > > ### Comment · Reviewer_2rv5 · 2022-11-13
> > > **basic idea valid, evaluation too weak**
> > >
> > > thank you for your extensive response. I can agree with many points made regarding the title, discussion of related work and and also that the mentioned "missing" references from CVPR and ECCV 22 should not be hold against this paper (but they are certainly helpful for the authors as another reviewer also mentioned some of them).
> > >
> > > Based on the additional clarifications provided, I think that the proposed analysis of CO by adding simple patterns which allow networks to solve the given task with simple features and to observe how these will change under AT and CO is interesting and results could give new insights. However, at its current state, I think that the paper is ready for publication for the following reasons:
> > >
> > > 1) the paper is failing to provide the reader with a strict line of argumentation, both in terms of writing as well of the presentation of the experimental results.
> > >
> > > 2) the paper actually fails for the most part to provide a qualitative analysis of how the learned features are changing under clear training, AT and CO. For example the qualitative results in Fig. are very interesting, but there is no quantitative analysis, e.g. in terms of a frequency analysis that shows how the attack patterns change from low to high frequency. The same holds for the non-linearity aspects.
> > >
> > > 3) results of attack pattern analysis, non-linearity and curvature, stand independently - it would be really beneficial to see quantitative results linked, i.e. in one plot over training
> > >
> > > 4) the provided indicators allow only an indirect observation of the features in terms of attack patterns and curvature of the optimization space - why are the authors omitting available direct measures like feature-map analysis (sparsity, frequency components ...) or Class activation Maps?
> > >
> > > 5) experiments are limited to one DCT pattern only. This probably provokes the net to learn low-frequency features in the clean training and shift towards high frequencies at CO (however the analysis is missing to show this - see 4) ). In this context it would be very interesting to see how the features would behave for other (i.e. high frequency) DCT patterns. If you would then see a shift towards low frequencies, it would be a strong indication that OC is indeed more than "just overfitting"
> > >
> > > I would like to encourage to the authors to pursue the core idea presented here. But in its current state, I will still vote against the acceptance of this paper.

---

> > > > ### Author Response · Authors · 2022-11-15
> > > > **Reply to new concerns (Part 1 of 2)**
> > > >
> > > > We thank the reviewer for engaging with us in a discussion and appreciate that they agree with many points of our previous response. In the following, we will address the new concerns raised by the reviewer, however, first we would like to make a general comment as we think there is a misunderstanding regarding the concept of features.
> > > >
> > > > ### General comment about features and overfitting
> > > >
> > > > In the reviewers’ initial CVPR references [1, 2], the authors study feature-maps of networks trained with various AT strategies. We are aware that the term feature-map or intermediate features is so common in deep learning that it is sometimes referred to as simply “network features” or even “features”. However, in our work, **when we talk about features we do not mean feature maps**. We talk about features in the “classic” machine learning sense, i.e.  any measurable signal in the input-space. This distinction is crucial to assess the “correctness” of our claims since **when we say that “we study features”, we are not missing important experiments simply because we did not study feature maps** as feature maps are a different entity and thus, a different research direction. However, as we mention in our point 2., we did perform experiments to quantitatively assess the predictive power of the learned feature maps and how they evolve, so we politely ask the reviewer to revisit this section.
> > > >
> > > > Similarly, we would like to mention again, that **despite the reviewer’s intuitions, CO is not “just overfitting”**. As we mentioned in our previous response, when CO happens in our experiments the network achieves both 100% train AND test accuracy, and hence has not overfitted in the classical sense. Therefore, regardless of the usefulness of our experiments, **we strongly believe that further studies in CO are necessary to understand this phenomenon, as no theory of classical overfitting can explain it**.
> > > >
> > > > ### New concerns raised by the reviewer
> > > >
> > > > #### **1. Writing and presentation inadequate**
> > > >
> > > > Thank you for the feedback, although we are a bit surprised to hear this from the reviewer as this was not mentioned in their initial comments. Furthermore, this seems to be in stark contrast to the feedback from all other reviewers: “the paper is well-written and supported by adequate graphics to demonstrate the author's findings” (F9cZ), “really easy and enjoyable to read” (27L3) and “overall it’s a solid and well written paper.” (GG1w). In any case, we would like to kindly ask the reviewer if they could provide further feedback on what specific parts of the paper they found hard to follow and we will try to clarify them and improve them in the draft based on the reviewer’s feedback.
> > > >
> > > > #### **2. Lack of quantitative analysis on learned features:**
> > > >
> > > > **We would like to point the reviewer to multiple experiments throughout our work that quantitatively assess which input features are learned.** Specifically, we would like to point the reviewer to Figure 3, in which we quantitatively analyze how important are different input features (i.e., injected amd original) for classification throughout training with different strategies.
> > > >
> > > > Similarly, in Figure 4 (right), we further quantify this notion and explore the capacity of the neural networks to extract useful feature maps to classify CIFAR-10. This is a quantitative and dynamic analysis of the learned features at the last layer, performed over different training runs.
> > > >
> > > > All these quantitative experiments indicate that prior to CO, the FGSM-AT seems to use input features from both CIFAR-10 and our injected set $\mathcal{V}$, however, after CO, the network focuses only on the injected features while still preserving a high FGSM accuracy. In later experiments, we link these observations to the non-linearity experiments (Figure 4 left) where we observe a curvature explosion (phase 2) that leads to CO.
> > > >
> > > > Finally, **regarding our analysis of attack patterns, following the reviewers suggestion, we have quantified our analysis and added two plots in Fig 18 in Appendix G**. Interestingly, as we can see, after CO, the FGSM attacks experience a rapid misalignment with the injected features and behave close to random vectors. Moreover, we also observe that their spectrum on the DCT components has a strong shift towards the high frequencies.  We thank the reviewer for the suggestion of these experiments that further confirm our observations and that we hope have served to alleviate some of the reviewer’s concerns.

---

> > > > > ### Author Response · Authors · 2022-11-15
> > > > > **Reply to new concerns (Part 2 of 2)**
> > > > >
> > > > > #### **3. Lack of quantitative results evolving over training:**
> > > > >
> > > > > **Again, we would like to refer the reviewer to different plots in our paper that show the evolution of different metrics throughout training**, thus allowing us to pinpoint and understand the dynamics of CO. Specifically, Figures 3, 4, 6, 14, 15, and 18, all show the evolution of metrics as the reviewer suggests.
> > > > >
> > > > > The reason we presented our results in different plots is to help digest all the information which otherwise would have appeared clattered and confusing. Still, all of the quantitative curves presented along training are comparable and serve to establish links between different quantities (e.g. feature analyses in Fig 3 with curvature in Fig 4 and Fig 6. Several additional plots can be found in Appendix regarding features (Fig 13), curvature (Fig 14, 15) and the recently added Fig 18 regarding the evolution of attack patterns.)
> > > > >
> > > > > #### **4. No feature map analysis:**
> > > > > As mentioned in our response to point 2., **we do in fact quantitatively study the evolution of the last layer features extracted by different training runs**. In particular, we assess how much are the extracted features useful to classify (unmodified) CIFAR images which further supports our findings in Fig 3.
> > > > >
> > > > > Given that our focus is on the input-space features (not on the feature maps) we respectfully disagree with the reviewer’s statement: “the provided indicators allow only an indirect observation of the features”. Precisely our intervention allows us to study and observe a particular set of input features that we know have caused CO. **Simply analysing feature maps in the context of CO would lead to different kinds of conclusions, which may be equally interesting, but would not have allowed a direct study of our observations**. Thus, we humbly believe that our experimental choice of injected features was the most sensible approach to study the effect that easy-to-learn but non-robust input features could have on adversarial training. We agree with the reviewer that further analyses of intermediate feature maps, especially linked to our injected features, could be interesting for future work. However, we do not think these are necessary to support the claims made in this paper.
> > > > >
> > > > > #### **5. Experiments are limited to one DCT pattern only.**
> > > > >
> > > > > **Again, we would like to point the reviewer to Figure 11 of the Appendix where we replicate our experiments using random vectors as injected signals rather than DCT components and observe the exact same behaviour as in Figure 1**. This suggests that CO is not simply a matter of low vs high frequency features. This is also corroborated by our low-pass experiments in Section 5 and the fact that the anti-aliasing layers of [3] do not work at large $\epsilon$.
> > > > >
> > > > > Moreover, as shown in Figure 3 and discussed in Section 4.1, after CO, the network trained with FGSM-AT keeps on using the injected features even more (injected dataset clean accuracy goes up but the accuracy on the unmodified CIFAR and the dataset with shuffled injected features drop) although the attacks get misaligned with these features. In this sense, the reviewers’ intuition “this probably provokes the net to learn low-frequency features in the clean training and shift towards high frequencies at CO” is not correct. The network keeps using the low-frequency injected features, but the FGSM attacks shift due to the curvature explosion. We have provided extensive quantitative evidence for this as discussed in our response to point 2.
> > > > >
> > > > > [1] Paul Gavrikov and Janis Keuper, CNN Filter DB: An Empirical Investigation of Trained Convolutional Filters, CVPR 2022
> > > > >
> > > > > [2] Paul Gavrikov and Janis Keuper, Adversarial Robustness through the Lens of Convolutional Filters, CVPR-W 2022
> > > > >
> > > > > [3] Julia Grabinski, Steffen Jung, Janis Keuper, and Margret Keuper. FrequencyLowCut Pooling - Plug & Play against Catastrophic Overfitting. ECCV, (October) 2022.

---

> > > > > ### Comment · Reviewer_2rv5 · 2022-11-15
> > > > > **features**
> > > > >
> > > > > let me split my response into into multiple threads - I think it is easier to discuss individual aspects separately...
> > > > >
> > > > > About features: it has become clear that the authors are referring to features as properties of the (image) input. However, the mappings learned by NN into the intermediate outputs (feature-maps) are representing what a NN is learning and reflect which of these features are actually used to form (robust) decisions. Hence, this can not be treated independently. The authors also rely on this aspect when using the curvature of the optimization space, which is dictated by the learned weights and thus also an indirect way to of analysis as feature-maps would be...
> > > > >
> > > > > Sidenote: [1,2] are not investigating feature-maps, but the learned filter-kernels...

---

> > > > > > ### Author Response · Authors · 2022-11-15
> > > > > > **Author response**
> > > > > >
> > > > > > In our work we do study which features are being learned by the network with two main experiments:
> > > > > >
> > > > > > 1. In Fig 3 we leverage different types of data intervention to understand how much is the network relying on the injected features vs other features present in CIFAR.
> > > > > >
> > > > > > 2. In Fig 4 we study the discriminative power of final layer feature maps to classify CIFAR.
> > > > > >
> > > > > > Although further analyses with the features could be performed, we would like to reiterate that we do not think they are necessary to support our claims. If the reviewer does not agree we do encourage the reviewer to explain what experiments they think our work is missing and for what reason.
> > > > > >
> > > > > > **Side note:** Just to clarify, as it is common in the adversarial robustness literature, we perform an analysis on the curvature of the loss with respect to the input, not the network parameters. In this regard, it is very unclear to us, how the properties of the learned filters would be able to intuitively explain anything about the geometry of the input loss landscape.

---

> > > > > ### Comment · Reviewer_2rv5 · 2022-11-15
> > > > > **overftting**
> > > > >
> > > > > Quote: "despite the reviewer’s intuitions, CO is not “just overfitting”
> > > > >
> > > > > Sorry, but I simply don't see that this is shown by any of the experiments in this paper or prior work. All observed indicators, from higher curvature to  the shift high frequency attack patterns and the proposed counter measures in form of regularization and augmentation are pointing in this direction. I'm not seeing any result that can not explained with overfitting to the attacks...
> > > > >
> > > > > However, I don't think that this actually hurts the value of the proposed method. A clear analysis based on the core idea of the paper would be very valuable... it is the way this analysis is conducted an presented what makes me reject this paper

---

> > > > > > ### Author Response · Authors · 2022-11-15
> > > > > > **Author response**
> > > > > >
> > > > > > We thank the reviewer for their open discussion, but we respectfully disagree with their claims. In particular, we would like to humbly ask the reviewer to explain the following results if CO was “just overfitting to the attack”:
> > > > > >
> > > > > > 1. In our experiments, when we inject features in the dataset, we observe that CO happens at an $\epsilon$ where it does not happen otherwise (i.e. $\beta = 0$ vs $\beta\approx\epsilon$). However, this same intervention does not cause CO when trained with PGD.
> > > > > >
> > > > > > 2. In (Andriushchenko and Flammarion, 2020), they show that if you project PGD perturbations to the corners of the $\ell_\infty$ ball (like the ones in FGSM), they DO NOT observe CO. This experiment led these authors to claim that CO is not just overfitting to the attacks.
> > > > > >
> > > > > > From our understanding, these experiments cannot be explained simply by considering CO as overfitting to the FGSM perturbations, but can be explained by our new insights. In fact, the results of Andriushchenko and Flammarion, 2020 triggered the spawn of many papers trying to explain CO beyond “overfitting to the attacks”.

---

> > > > > > > ### Comment · Reviewer_2rv5 · 2022-11-16
> > > > > > > **Overfitting**
> > > > > > >
> > > > > > > regarding 1: in your experiments, the capacity of the used NN is constant (NN architecture does not change). Injecting simple to classify features, most likely results in a network that is using much less of its learning capacity to solve this task than it would need for the original problem. Hence, it has more capacity left for overfitting. [1] showed that non-bust models only use a portion of their capacity (by analyzing the learned weights). Therefore, again it would be important to extend your analysis to intrinsic properties of the models...
> > > > > > >
> > > > > > > Regarding PGD: here it is known that AT with PGD is much less prune to CO in general, so again more experiments would be required
> > > > > > >
> > > > > > > regarding 2: I'm aware of the work by (Andriushchenko and Flammarion, 2020), while this provides interesting insides, I don't agree that one generally could conclude that CO in not overfitting. Besides, the key conclusion should be provided  by he paper, not prior work (novelty). Again, I don't think that this aspect is harming the paper in any way, if CO is indeed a form of overfitting

---

> > > > > ### Comment · Reviewer_2rv5 · 2022-11-15
> > > > > **presentation inadequate**
> > > > >
> > > > > Quote: "although we are a bit surprised to hear this from the reviewer as this was not mentioned in their initial comments"
> > > > >
> > > > > well, the initial review assumed that the basic approach is not valid. After consideration of the author response and putting quite a lot of work into piecing individual results together, I changed my opinion regarding the core idea of the paper. Still, the line of argumentation, choice of evaluation methods and presentation of the results is very hard to follow. It is a strong point against a paper if it needs long explanations and the study of 14 pages of supp. materials

---

> > > > > > ### Author Response · Authors · 2022-11-15
> > > > > > **Author response**
> > > > > >
> > > > > > We thank the reviewer for their effort in revisiting the paper after our initial comments and we appreciate that afterwards, they changed their mind and found the core idea of the paper interesting, and our subsequent experiments to be in the right direction.
> > > > > >
> > > > > > Again, we humbly would like to invite the reviewer to read the other reviewers comments (especially the strengths and summary of reviews). All other reviewers agree that the paper is well written and the presentation adequate, and claim that our observations are well-supported by empirical and theoretical justifications. As it seems that the reviewer strongly disagrees with those opinions, we would like to encourage the reviewer to please explain in which way our explanations are not clear and how they think our analysis could be made more thorough. Precisely, the additional 14 pages of Appendix of our work should be seen as a clear sign of our effort to be as thorough and rigorous as possible in our exposition, as we replicated our experiments (with the same findings) in a multitude of settings.

---

> > > > > > > ### Comment · Reviewer_2rv5 · 2022-11-16
> > > > > > > **other reviews and quality of presentation**
> > > > > > >
> > > > > > > Unfortunately, the other reviewers have not joined our constructive discussion. I would find it very helpful there would be some feedback...
> > > > > > >
> > > > > > > So far, the other reviewers voted:  3 - with some of the same arguments as I have, and twice 6. The later reviews are quite brief and the low reviewer confidence, the basically neutral score as well as the missing participation in our discussion make it look like there is no one who rely supports this paper. In contrast, I actually think that the proposed Idea has the potential to become a good A* paper, just that the current state of presentation and limited evaluation is not at A* level. If this was journal, I would vote for a major revision. Unfortunately, in a conference this results in a reject and resubmit.

---

> > > > > > > > ### Author Response · Authors · 2022-11-16
> > > > > > > > **Author response**
> > > > > > > >
> > > > > > > > Thank you for your personal assessment of the other reviews. To the best of our knowledge, however, the shared concern with Reviewer F9cZ is that we did not cite the concurrent work of (Grabinski et al. 2022). As we mentioned in our general comment to all reviewers:
> > > > > > > >
> > > > > > > > 1. Not citing this work should not be grounds for rejection.
> > > > > > > >
> > > > > > > > 2. We have still cited this work in the revision of our manuscript.
> > > > > > > >
> > > > > > > > Moreover, [as the reviewer previously stated](https://openreview.net/forum?id=n0okuXMlI7V&noteId=_KEdVnWGSy): "the mentioned 'missing' references from CVPR and ECCV 22 should not be hold against this paper".
> > > > > > > >
> > > > > > > > Finally, we reiterate that in the words of the other three reviewers:  “the paper is well-written and supported by adequate graphics to demonstrate the author's findings” (F9cZ), “really easy and enjoyable to read” (27L3) and “overall it’s a solid and well written paper. [...] The way the authors design experiments is novel. Both empirical results and theoretical justifications are provided." (GG1w)

---

> > > > > ### Comment · Reviewer_2rv5 · 2022-11-15
> > > > > **quantitative analysis on learned features**
> > > > >
> > > > > thanks for fig. 18. This points in the direction I think would be a very valuable line of analysis. Still I think that the analysis could much better structured, i.e. combining different indicators in one plot over epochs with marked OC.

---

> > > > > > ### Author Response · Authors · 2022-11-15
> > > > > > **Author response**
> > > > > >
> > > > > > As we mention in our post [Reply to new concerns (Part 2 of 2)](https://openreview.net/forum?id=n0okuXMlI7V&noteId=uwEixHSDbi6) (specifically in point 3) all the different aspects that we study during training (e.g. curvature, learned features, etc.) are comparable among each other. In this sense, we respectfully, but strongly, disagree with the reviewer that a single plot showing several different metrics with a different y-axis would be better to interpret. Whenever it was necessary for the exposition, we clearly denoted both in the text and the figures the stages in which CO happens, and made sure to always provide results for the same run (although we also show aggregate metrics for multiple runs) in all our plots.
> > > > > >
> > > > > > In any case, following the reviewers’ suggestion, we have now modified Fig. 18 and also marked here when CO happens in this figure.
> > > > > >
> > > > > > Although we appreciate these comments, we would like to state that we do not think that such aesthetic choices should justify that a paper is rejected, especially given the fact that all other reviewers found the paper well-written and presented.

---

### Official Review · Reviewer_27L3 · 2022-11-03

**Confidence:** 3
**Correctness:** 4
**Technical Novelty And Significance:** 3
**Empirical Novelty And Significance:** 3
**Recommendation:** 6

**Clarity, Quality, Novelty And Reproducibility:**

The paper is well-written and insightful. In general, the paper is original and timely as it tackles an important problem. The proposed approach directly extends the previous work on “Adversarial Examples Are Not Bugs, They Are Features”. Yet, the proposed approach is novel as it addresses the problem of catastrophic overfitting. At the time of the review, the authors haven’t released the source code, which makes it more difficult to reproduce the results. The authors should try to release the source code of the paper.

**Strength And Weaknesses:**

### Strengths

- A novel approach to understanding catastrophic overfitting based on controlled manipulations of the dataset, which provides interesting insights into the problem.
- A novel explanation of catastrophic overfitting that can be used to guide the development of new defense methods.
- Detailed extensive empirical analysis of catastrophic overfitting under different dataset manipulations.
- The paper is really easy and enjoyable to read.

### Weaknesses

- Technical novelty is limited, all results are very intuitive.
- At the time of the review, the author’s code is not available, which will be hopefully addressed in the revision phase.

**Summary Of The Paper:**

The authors attempt to understand catastrophic overfitting in adversarial training with an FGSM adversary. To analyze catastrophic overfitting, they proposed to induce it using controlled manipulations of the dataset technique. Depending on the magnitude of the data perturbations, they observed that it is possible to induce catastrophic overfitting to any model given sufficiently large dataset perturbation. The authors discussed the implications of this finding and provided further insights into catastrophic overfitting.

**Summary Of The Review:**

This paper provides novel insights into catastrophic overfitting with fast adversarial training. Understanding catastrophic overfitting is a very important problem in adversarial machine learning and addressing it is extremely important. To this end, the authors proposed a novel method to induce catastrophic overfitting in controlled settings, provided possible explanations of this phenomenon, and suggested a few directions to reduce catastrophic overfitting in practice. Overall, the paper is interesting and enjoyable to read and makes few contributions.

---

> ### Author Response · Authors · 2022-11-10
> **Response to reviewer 27L3**
>
> We thank the reviewer 27L3 for their effort and valuable comments. We appreciate that they find our paper well-written, insightful, and our approach to studying CO along with our explanations novel. We would like to point the reviewer to [our general message to all reviewers](https://openreview.net/forum?id=n0okuXMlI7V&noteId=e1vUYSktrZY). In what follows we address the particular concerns of this reviewer.
>
> > Missing code.
>
> As we mentioned in the general message to all reviewers, we have just released the code to reproduce our experiments as suggested by the reviewer and we will publish it in a github repository once the work is deanonymized.
>
> > Technical novelty is limited, all results are very intuitive.
>
> We respectfully disagree. The fact that CO can be induced at lower $\epsilon$ by injecting easy features, and the dependency of this phenomenon on $\beta$ is, in our humble opinion, a rather surprising finding, which could not have been intuitively explained by prior work. On the other hand, **if the reviewer 27L3 found our explanations clear and intuitive, we humbly argue this should be a strength of our work and not a weakness**. In this regard, we would like to respectfully ask the reviewer to reconsider their score. Our results provide a very novel explanation of CO, which although it may appear intuitive in hindsight, it was not obvious a priori.
>
> Since we have addressed the reviewer's main concern by releasing our code, and argued that clear and intuitive exposition is not a weakness, we respectfully ask the reviewer to consider increasing their score.

---

### Official Review · Reviewer_F9cZ · 2022-11-03

**Confidence:** 4
**Correctness:** 3
**Technical Novelty And Significance:** 2
**Empirical Novelty And Significance:** 3
**Recommendation:** 5

**Clarity, Quality, Novelty And Reproducibility:**

- **Clarity and Quality:** The paper is well-written and supported by adequate graphics to demonstrate the author's findings.
-  **Novelty:** The authors missed one prior work which shows a different perspective on CO. Thus, I would suggest the authors to include this method in their analysis part as well as the insights on CO in their discussion.
- **Reproducibility:** The code for reproducing the results of the paper should be publicly available to encourage future research on the explainability of CNNs.


**Strength And Weaknesses:**

**Strength:**
- The paper is well-written and supported by adequate graphics to demonstrate the author's findings.
- The authors provide an empirical analysis of the learning preferences of CNNs under different training settings (standard training, FGSM AT and PGD AT)
- The approach of injecting simple features for the analysis of CO is interesting and can be extended for future explainability studies on CNNs

**Weakness:**

- Prior work/extensiveness: the authors investigated prior work that prevents CO, however, missed prior work which includes additional insights into why CO happens:
[0] https://arxiv.org/abs/2204.00491 (ECCV 2022): showing that CO is correlated with a vast increase in aliasing and preventing CO by using an aliasing-free downsampling layer

- The authors conducted their empirical study only on low-resolution datasets and for small networks. It would be interesting to see if their approach also applies to high-resolution data and bigger networks.
- The authors did not provide any code for reproducibility.


**Summary Of The Paper:**

In this paper, the authors investigate the phenomenon of catastrophic overfitting (CO) which occurs during FGSM adversarial training (AT). CO describes the “overfitting” of the model to simple adversaries (FGSM) during a certain amount of training epochs while losing robustness against more complex adversaries like PGD.

The authors show empirically:

1. that it is possible to induce catastrophic overfitting on smaller \epsilon values (where \epsilon is the hyperparameter to determine the strength of the adversarial attack) than before by adding simple discriminative features to the dataset. These injected features are added onto the original image scaled with a hyperparameter $\beta$
2. if $\beta << \epsilon$ the model behaves normal
3. if $\beta >> \epsilon$ the robust accuracy is high as the model can rely on the simple features
4. if $\beta \approx \epsilon$ they can induce CO at \epsilon values at which the model on the clean data would not suffer from CO

The part where $\beta \geq \epsilon$ is further exploited to analyse the network's training behavior, especially at the point of CO. Further, the authors empirically show:

1. that standard training and FGSM training after CO highly rely on the simple features while PGD training forces the model to learn more robust features
2. that the curvature of the model’s loss landscape explodes after CO

The authors show empirically that models favour easy-to-learn features when no robustness constraints are considered. Further, they claim that a network that needs to learn different features needs to increase its non-linearity and thus opens a shortcut to breaking FGSM with CO.

Further, the authors analyse two methods which can prevent CO (GradAlign and N-FGSM). They empirically show that these methods can keep low curvature of the loss landscape thus preventing CO. Additionally, the authors try to use a lowpass filter on top of the data but could not prevent CO for large $\epsilon$ values.


**Summary Of The Review:**

The contribution of this paper is a simple, yet interesting empirical analysis of CO. However, the authors missed a new perspective from previous work. Thus, I would like to encourage the authors to include this perspective in their analysis. Further, it would be interesting to see if the presented results hold for high-resolution datasets and bigger networks.
Additionally, the authors should provide their code for reproducibility and enablement of future research on the analysis of the explainability of CNNs.

Generally, the suggested idea and findings are interesting and worse publishing. However, I can not accept the paper for now due to my concerns mentioned above. I highly encourage the authors to address these concerns such that I can increase my score.

---

> ### Author Response · Authors · 2022-11-10
> **Response to reviewer F9cZ**
>
> We thank the reviewer for their effort and valuable comments. We appreciate that reviewer F9cZ  finds our paper well-written and our ideas and findings worth publishing. We would like to point the reviewer to [our general message above](https://openreview.net/forum?id=n0okuXMlI7V&noteId=e1vUYSktrZY) where we address their main concerns:
> 1. The discussion of concurrent work presented two weeks ago at ECCV 2022 [1].
> 2. The release of our code to reproduce experiments.
>
> To summarise, we do think that [1] is an interesting concurrent work. However, note that our experiments in Section 5 and the new Appendix I show that aliasing due to high-frequency components is not enough to fully characterise CO. Our findings in this work are complementary to those in [1] and can help future research in the field to obtain robust models more efficiently. Similarly, our code has been released to help future research.
>
> > More experiments with different setups.
>
> Regarding the need to experiment with larger networks, we politely refer the reviewer to our Appendix D where we present many extensive thorough ablations and analyses of our experiments. Specifically in Appendix D.4, we include a larger architecture WideResNet28x10 and observe that our injected features also induce CO. Moreover, we also replicate our findings on two other datasets, using different types of injected features and other $\ell_p$ norms. Overall, our experiments already amount to more than 2,000 GPU-hours, which was the very limit of our compute ability. In this regard, we did not perform experiments on high-resolution datasets due to the much higher computational cost as our experimental setup requires running the full training loop for multiple versions of the dataset (varying $\beta$) multiple times. Note that for a given $\beta$ curve (where we vary the strength of our injected features), we adversarially train and test 15 conditions, $\times$ 3 seeds for rigorousness, which is very demanding.
> All in all, we believe that all our experiments have provided ample empirical evidence to convincingly show our main claim that the interaction between different types of features (robust/non-robust and easy/hard) is key to explain the causes of CO. **Our observations are consistent across datasets and networks and generalise to other attack norms and designs of injected features.**
>
> We hope we have addressed the reviewer's concerns so we politely ask the reviewer to consider increasing their score.
>
> [1] Julia Grabinski, Steffen Jung, Janis Keuper, and Margret Keuper. FrequencyLowCut Pooling - Plug & Play against Catastrophic Overfitting. European Conference on Computer Vision (ECCV), October 2022.

---

> > ### Comment · Reviewer_F9cZ · 2022-11-16
> > **related work and high-resolution dataset**
> >
> > I thank the authors for their response and appreciate that they are willing to publish the code for reproducibility.
> > Coming to the two points addressed:
> >
> > 1. Concurrent work
> >
> > While you evaluate data manipulation, [1] did not manipulate the data itself. Instead, they improve the downsampling layer. Thus, I do not believe that simple data manipulation results in the same performance as changing the network components. Hence, I’m not convinced that your findings with increased epsilon values on data manipulation transfer 1:1 to the work in [1].
> >
> > 2. High-resolution dataset
> >
> > I did see those results in the appendix and appreciate that you evaluated your approach on other low-resolution datasets. However, I’m concerned that your method does not expand to high-resolution datasets as these mostly include more details (= more complex/robust features?) which are used for training. I’m aware of the huge size of for example ImageNet, nevertheless, there exist smaller versions of ImageNet (e.g., ImageNet100 or even smaller versions with slightly lower resolution ImageNet64x64). Hence, I would encourage the authors to use these datasets to give at least a hint if their method is also valid for high-resolution datasets.
> >
> > Consequently, I will keep my original score and encourage the authors to include high-resolution datasets. Further, I would like to motivate the authors to carefully read through the related work and include their perspectives (also the related work mentioned by 2rv5 seems a vulnerable perspective).

---

> > > ### Author Response · Authors · 2022-11-17
> > > **New results on ImageNet100 and TinyImageNet**
> > >
> > > We would like to thank reviewer F9cZ for engaging in the discussion. In the following we address their new comments:
> > >
> > > > Concurrent work
> > >
> > > First, we would like to re-emphasise that **[1] is contemporaneous work** and as such, the reviewing guidelines state that  it need not be discussed in our work. For the sake of completeness, however, and per the reviewer's suggestion, we still discussed and performed experiments regarding [1] in Section 5, and Appendix I of our work.
> > >
> > > Specifically, we took the public code of [1] and re-tested it by training on CIFAR-10 (without injected features) at $\epsilon=16/255$. To our surprise, we found that in that regime, the novel pooling layers described in [1] do not prevent CO. This led us to conclude that **the new interesting aliasing perspective of [1] cannot fully explain the onset of CO**. Moreover, as suggested by the reviewer, in our text we draw parallels between this novel pooling layer (that removes high frequencies before downsampling) with our low-pass experiments, as they both share many commonalities.
> > >
> > > > High-resolution dataset
> > >
> > > We are happy to share with the reviewer that **we have been able to show results for Imagenet-100 following their initial suggestion**. Moreover, **we have also included additional experiments with Tiny Imagenet** following the recent comment of the reviewer. In both cases results can be found in Appendix D.1, but we copy the results for ImageNet-100 here for ease of reference:
> > >
> > > |                       | Clean accuracy | FGSM accuracy | Robust accuracy |
> > > |:---------------------:|:--------------:|:-------------:|:---------------:|
> > > | $\beta=\frac{0}{255}$ |      71.0%     |     49.1%     |      45.6%      |
> > > | $\beta=\frac{2}{255}$ |      70.9%     |     51.16%    |      45.1%      |
> > > | $\beta=\frac{4}{255}$ |      74.2%     |     52.56%    |      42.1%      |
> > > | $\beta=\frac{5}{255}$ |      100%      |     99.8%     |       0.0%      |
> > > | $\beta=\frac{6}{255}$ |      100%      |     98.0%     |       1.9%      |
> > >
> > > Overall, we have been able to observe a similar pattern in both datasets as to the one presented in the main paper for CIFAR-10 (and previously shown for CIFAR-100 and SVHN, in the Appendix), i.e. as we increase $\beta$ FGSM-AT has three regimes:
> > >
> > > 1. $\beta<\epsilon$: FGSM-AT behaves as though there was no injected feature.
> > >
> > > 2. $\beta\approx\epsilon$: We observe that the FGSM test accuracy and clean accuracy suddenly go up, while the Robust accuracy drops significantly.
> > >
> > > 3. $\beta\gg\epsilon$: The injected features are already robust, so the Robust accuracy also goes up, greatly surpassing the values of $\beta=0$ (results for larger $\beta$ are still running for ImageNet-100).
> > >
> > > With the inclusion of these experiments (high resolution and ECCV concurrent work), as well the released code to reproduce results, **we believe we have addressed all of the reviewers' concerns**, therefore, we respectfully ask the reviewer to increase their score, as they had said in their initial comment.

---

> > > > ### Comment · Area_Chair_QF6q · 2022-11-25
> > > > **results on high resolution datasets**
> > > >
> > > > Dear reviewer F9cZ,
> > > >
> > > > did you get a chance to have a look at the results on high resolution data on ImageNet-100 provided above?
> > > > Do they address your concerns on the validity for high resolution data?
> > > >
> > > > Best regards,
> > > > AC

---

> > > > > ### Comment · Reviewer_F9cZ · 2022-11-25
> > > > > **results on high resolution datasets**
> > > > >
> > > > > I appreciate that the authors followed my advice and started their experiments on higher-resolution datasets. However, the full results are still pending (as mentioned by the authors).
> > > > > Thus, I would like to rise my score to a borderline rejection but need to agree with Reviewer 2rv5, that the paper would need to get a major revision to be accepted.
> > > > > However, if other Reviewers argue to accept the paper I'm not strongly against accepting it as the general idea is interesting and can be extended for future explainability studies on CNNs (as already mentioned above).

---

> > > > > > ### Author Response · Authors · 2022-11-25
> > > > > > **Missing results**
> > > > > >
> > > > > > Thank you very much for raising the score. Below you can find the results for the larger $\beta$s the reviewer is missing which follow the same pattern observed on other datasets (including the new TinyImageNet). We will upload results for $\beta=12/255$ and $\beta=14/255$ tomorrow, however, we respectfully ask the reviewer to be mindful of the short time of the rebuttal period, and reassess if these two additional points are crucial to support our claims.
> > > > > >
> > > > > > |                        | Clean accuracy | FGSM accuracy | Robust accuracy |
> > > > > > |:----------------------:|:--------------:|:-------------:|:---------------:|
> > > > > > |  $\beta=\frac{0}{255}$ |      71.0%     |     49.1%     |      45.6%      |
> > > > > > |  $\beta=\frac{2}{255}$ |      70.9%     |     51.16%    |      45.1%      |
> > > > > > | $\beta=\frac{4}{255}$  |      74.2%     |     52.56%    |      42.1%      |
> > > > > > | $\beta=\frac{5}{255}$  |      100%      |     99.8%     |       0.0%      |
> > > > > > | $\beta=\frac{6}{255}$  |      100%      |      100%     |       1.9%      |
> > > > > > | $\beta=\frac{8}{255}$  |      100%      |      100%     |      34.3%      |
> > > > > > | $\beta=\frac{10}{255}$ |      100%      |      100%     |      59.4%      |
> > > > > >
> > > > > > As for the proposed major revision, could the reviewer please explain what needs to be revised in order to be accepted? We will be happy to implement this in the final version. In their previous comment, this reviewer said "The paper is well-written and supported by adequate graphics to demonstrate the author's findings."

---

> > > > > > > ### Author Response · Authors · 2022-11-26
> > > > > > > **Final results on ImageNeet100**
> > > > > > >
> > > > > > > As promised, here are the complete results of our experiments on ImageNet100 where we can clearly observe the three $\beta$ regimes described in our work. We hope the reviewer is satisfied with the inclusion of these complete results.
> > > > > > >
> > > > > > > |                        | Clean accuracy | FGSM accuracy | Robust accuracy |
> > > > > > > |:----------------------:|:--------------:|:-------------:|:---------------:|
> > > > > > > |  $\beta=\frac{0}{255}$ |      71.0%     |     49.1%     |      45.6%      |
> > > > > > > |  $\beta=\frac{2}{255}$ |      70.9%     |     51.16%    |      45.1%      |
> > > > > > > | $\beta=\frac{4}{255}$  |      74.2%     |     52.56%    |      42.1%      |
> > > > > > > | $\beta=\frac{5}{255}$  |      100%      |     99.8%     |       0.0%      |
> > > > > > > | $\beta=\frac{6}{255}$  |      100%      |      100%     |       1.9%      |
> > > > > > > | $\beta=\frac{8}{255}$  |      100%      |      100%     |      34.3%      |
> > > > > > > | $\beta=\frac{10}{255}$ |      100%      |      100%     |      59.4%      |
> > > > > > > | $\beta=\frac{12}{255}$ |      100%      |      100%     |      83.8%      |
> > > > > > > | $\beta=\frac{14}{255}$ |      100%      |      100%     |      93.8%      |

---

### Author Response · Authors · 2022-11-10
**General comment to all reviewers**

We thank all the reviewers for their valuable feedback and comments. We are delighted the reviewers find our work well-written ([F9cZ](https://openreview.net/forum?id=n0okuXMlI7V&noteId=qqSGXaqSXa), [GG1w](https://openreview.net/forum?id=n0okuXMlI7V&noteId=cm65VkUqsbT)) and enjoyable to read ([27L3](https://openreview.net/forum?id=n0okuXMlI7V&noteId=SGitGul_aq)), our intervention approach interesting for future studies ([F9cZ](https://openreview.net/forum?id=n0okuXMlI7V&noteId=qqSGXaqSXa)), our explanation of catastrophic overfitting novel ([GG1w](https://openreview.net/forum?id=n0okuXMlI7V&noteId=cm65VkUqsbT), [27L3](https://openreview.net/forum?id=n0okuXMlI7V&noteId=SGitGul_aq)), and our experiments detailed and extensive ([27L3](https://openreview.net/forum?id=n0okuXMlI7V&noteId=SGitGul_aq)). Similarly,  they mention that the questions we addressed are important ([2rv5](https://openreview.net/forum?id=n0okuXMlI7V&noteId=q-SKnHh1Ad)), our theory for onset of CO is both empirically and theoretically justified ([GG1w](https://openreview.net/forum?id=n0okuXMlI7V&noteId=cm65VkUqsbT)); and the related work well cited and discussed ([GG1w](https://openreview.net/forum?id=n0okuXMlI7V&noteId=cm65VkUqsbT)).

The main concerns brought up by the reviewers are the following:

1. **[Concurrent work]** Our work misses a discussion with respect to a concurrent work [1] (presented two weeks ago in ECCV 2022).
2. **[Code Release]** We did not release our code yet.

We have addressed these concerns in our new draft (in red), in which we have adequately cited this concurrent work, and uploaded our code as new supplementary material.

To favour an open discussion, we make a few more comments regarding these concerns below (more individualised comments are given to each reviewer):

> Missing discussion about contemporaneous work [1].

We thank the reviewers ([F9cZ](https://openreview.net/forum?id=n0okuXMlI7V&noteId=qqSGXaqSXa), [2rv5](https://openreview.net/forum?id=n0okuXMlI7V&noteId=q-SKnHh1Ad))  for bringing this relevant paper to our attention. This is indeed an interesting work, which can be linked to some of our experiments in Section 5. However, we would like to note that this paper was officially presented just two weeks ago in ECCV2022, while the submission deadline to ICLR 2023 was one month before that. In this regard, we would like to emphasise that as **this is contemporaneous work**, and as clearly specified by [the reviewer guidelines](https://iclr.cc/Conferences/2023/ReviewerGuide#faq), **not citing or discussing this work should not be grounds for rejection**.

In any case, we have **still highlighted the link to this work in our new draft**. In particular, we note that our low-pass experiments in Section 5 are very similar in spirit to the FrequencyLowCut Pooling proposed in [1]. In this sense, both approaches support our main claim that the features of the data have an important role in CO. In our experiments, we found that applying a low-pass filter on the input seems to avoid CO for moderate $\epsilon$ perturbation radii (i.e. 8/255), and does not work for larger $\epsilon$ (i.e. 16/255). This led us to conclude that the existence of high-frequency non-robust features cannot fully explain CO.

To further confirm this, as suggested by the reviewers, we have retested this claim using the public code from [1]. Interestingly, we find  at $\epsilon=16/255$, the proposed FLC-Pooling layers do not prevent CO (see new Appendix I). This result was not tested in [1], but based on this finding we confirm that the existence of non-robust high-frequency features (or their potential aliasing artefacts due to downsampling) are indeed not enough to explain the causes of CO.

> No code released yet.

We are happy to share our code which has been attached as a .zip file in supplementary materials and, as was always our intention, we will publish it in a github repository once the work is deanonymized.

[1] Julia Grabinski, Steffen Jung, Janis Keuper, and Margret Keuper. FrequencyLowCut Pooling - Plug & Play against Catastrophic Overfitting. European Conference on Computer Vision (ECCV), October 2022.

---

### Author Response · Authors · 2022-11-15
**Second paper update**

Dear reviewers,

we would like to inform you that after a constructive exchange with [Reviewer 2rv5](https://openreview.net/forum?id=n0okuXMlI7V&noteId=WuyE0dswZv0) we have updated our manuscript to include a new Figure 18. in which we quantify the missalignment of FGSM perturbations before and after CO.

These quantitative results corroborate our previous observations, that before CO, FGSM perturbations are pointing towards meaningful predictive features, while after CO, although we know the network still uses the injected features (see Fig. 3) the FGSM perturbations suddenly point in a different direction.

Thank you very much.

---

### Author Response · Authors · 2022-11-17
**Third paper update**

Dear reviewers,

we would like to inform you that after a constructive exchange with [Reviewer F9cZ](https://openreview.net/forum?id=n0okuXMlI7V&noteId=PSfc2vUPyC) we have updated our manuscript to include new results on high resolution datasets: ImageNet-100 and TinyImageNet, where we are also able to induce CO by injecting simple features.

These results further corroborate our previous observations.

Thank you very much.

---

### Author Response · Authors · 2022-12-06
**Gentle nudge to reviewers**

Dear reviewers and AC,

Now that the discussion period is coming to an end, we would like to use this forum one last time to kindly ask the “two rather positive reviewers” (in the words of the AC) to voice their support for the paper once again.

There has been a very productive debate in this discussion period, after which the two more negative reviewers have finally decided to increase their score and openly stated they do not oppose an acceptance. However, it seems that the final decision of the paper will heavily rely on the engagement of the two positive reviewers. In this regard, and to facilitate their involvement in the discussion we would like to use this opportunity to make a summary of the discussion and updates to our work:

- We have discussed and referenced the concurrent work [1] demanded by reviewer F9cZ and 2rv5 in our new manuscript.
- We have clearly replicated our findings on high-resolution datasets (ImageNet-100 and TinyImageNet) as demanded by reviewer F9cZ.
- We have attached our code for full reproducibility as directed by reviewers 27L3 and F9cZ.
- We have included two new plots quantifying the alignment of the FGSM perturbations with the injected features and Fourier basis as requested by reviewer 2rv5.
- The only remaining new concern of 2rv5 and F9cZ seems to be a certain lack of clarity in parts of our text. This opposes the assessment of 27L3 (“really easy and enjoyable to read.”), GG1w (“overall it’s a solid and well written paper.”) and the original review of F9cZ (“the paper is well-written and supported by adequate graphics to demonstrate the author's findings”).   While the reviewers did not provide specific feedback on which editing they would like to see in this regard, we would be totally open to make these text changes in the camera-ready of our work and we hope that this will not be a reason for rejection of the paper.

We thank the reviewers and AC once again for the active conversation, and we hope that we can use these final days of the discussion period to address any remaining concerns.

The authors

---

### Decision · Program_Chairs · 2023-01-20

**Decision:**

Reject

**Justification For Why Not Higher Score:**

While the paper presents an interesting attempt to study CO, the current version lacks clarity in the line of argumentation. The paper needs a revision to discuss details and conclusions in a satisfying way.

**Justification For Why Not Lower Score:**

N/A

**Metareview: Summary, Strengths And Weaknesses:**

The paper presents a novel approach for the investigation of the "catastrophic overfitting" (CO) phenomenon, observed during adversarial training of CNNs. The authors propose to add trivial features based on DCT base functions in order to induce and study CO.

Initially, the paper received 4 reviews of which two reviewers with lower confidence gave a "marginally above threshold" rating and two reviewers with high confidences voted for a "reject", initially questioning the validity of the approach and missing citations.

While the authors were able to address many concerns in the following intense and productive discussion phase, both "reject" reviewers did not raise their scores above "marginally below threshold" while the other two reviewers remained inactive. A final attempt of the AC to clarify the votes in an online meeting with all reviewers did not result in a clear commitment towards acceptance from any reviewer.

The AC agrees with all reviews that the presented method is indeed a valid and interesting approach towards the understanding and prevention of CO. However, the discussion also showed that the paper currently is not ready for acceptance. The main remaining points are:

a) the line of argumentation and the presentation of experimental results needs to be clearer

b) more experiments, including ablations are needed to show the full potential of the approach

c) discussion and conclusions should give a better interpretation of the results, their impact on towards an understanding of CO and possible counter measures